# ROBUST REINFORCEMENT LEARNING WITH STRUCTURED ADVERSARIAL ENSEMBLE

## ABSTRACT

Although reinforcement learning (RL) is considered the gold standard for policy design, it may not always provide a robust solution in various scenarios. This can result in severe performance degradation when the environment is exposed to potential disturbances. Adversarial training using a two-player max-min game has been proven effective in enhancing the robustness of RL agents. However, we observe two severe problems pertaining to this approach: (*i*) the potential *over-optimism* caused by the difficulty of the inner optimization problem, and (*ii*) the potential *over-pessimism* caused by the selection of a candidate adversary set that may include unlikely scenarios. To this end, we extend the two-player game by introducing an adversarial ensemble, which involves a group of adversaries. We theoretically establish that an adversarial ensemble can efficiently and effectively obtain improved solutions to the inner optimization problem, alleviating the over-optimism. Then we address the over-pessimism by replacing the worst-case performance in the inner optimization with the average performance over the worst-$k$ adversaries. Our proposed algorithm significantly outperforms other robust RL algorithms that fail to address these two problems, corroborating the importance of the identified problems. Extensive experimental results demonstrate that the proposed algorithm consistently generate policies with enhanced robustness.

## 1 INTRODUCTION

Deep reinforcement learning (RL) has shown its success toward synthesizing optimal strategies over environments with complex underlying dynamics (Arulkumaran et al., 2017; Vinyals et al., 2019; Ibarz et al., 2021; Gao et al., 2022). However, given the large parameter search space under the function approximation schema and the limited scale of exploration over the state-action space during training due to sophisticated dynamics and environmental stochasticity (Shen et al., 2020), limited performance guarantees can be provided for the resulting policies. Consequently, there are often concerns regarding the robustness of RL (Pinto et al., 2017), i.e., whether RL policies can perform consistently well under unforeseeable external disturbances applied to the agent upon deployment. One framework that has been proven to effectively enhance the robustness of the RL agents is robustness through adversarial training (Gu et al., 2019; Kamalaruban et al., 2020; Pattanaik et al., 2017; Pinto et al., 2017; Vinitsky et al., 2020; Zhang et al., 2021). In this framework, the RL agent is assumed to share the environment with a hostile agent (adversary). The adversary takes actions to disturb the environment and/or the RL agent directly so that the cumulative reward received by the RL agent is minimized. Formulated as a max-min optimization problem, this framework optimizes the *worst-case* performance of RL agents under a pre-defined set of disturbance.

Despite these strengths of robustness through adversarial training, we observe two severe challenges pertaining to this approach. The first challenge is the *over-optimism* caused by the difficulty of the inner optimization problem. Without a closed-form solution, the optimal solution is approximated by a first-order method such as gradient descent that can be trapped in local optimum with high probability, resulting in an over-optimistic estimation of of the worst case performance. The second challenge is the *over-pessimism* caused by the selection of a candidate adversary set that may include unlikely scenarios. In most practical real-world scenarios it is often challenging, if not fully unfeasible, to have complete knowledge (e.g., probabilities of specific actions) of the environmental disturbances or the potential adversarial attacks. Consequently, most approaches only consider simple restrictions on the opponent's actions, such as the norm of the parameter or the entropy of the policy, leading to a

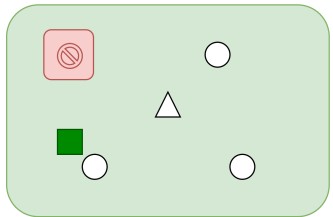 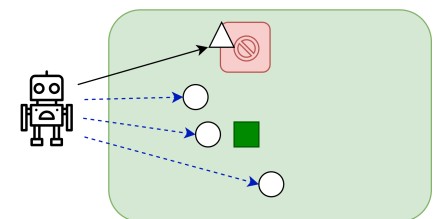

Figure 1: Motivation for an adversarial ensemble. The green regions represent the pre-defined set of adversaries. In real life applications it is challenging to choose the set of adversaries with high precision, leading to an adversary set with irrelevant or unlikely scenarios (denoted by the red square region). The triangle represents the single adversary in the regular adversarial training; the circles represent the extra adversaries in the adversarial ensemble. **Left**: The green square is the optimal adversary in the iterations of adversarial training. Compared to a single adversary, an ensemble can better approximate the optimal adversary, thus better approximating the inner optimization problem. **Right:** During the adversarial training, the protagonist can be diverted and become over-conservative if the single adversary steps into irrelevant regions. An ensemble can relieve this by distributing the attention of the protagonist from the irrelevant worst case to other cases.

candidate adversary set that lacks precision and thus is too broad. This can result in a over-conservative agent under the scheme of worst-case optimization. For instance, when training a controller for a helicopter, if the adversary is allowed to modify the environmental parameters to some physically unfeasible values, the agent must sacrifice its performance in real-world scenarios to improve its performance in these unlikely environments so that the worst-case performance is optimized.

To this end, we extend the two-player max-min game by introducing a structured adversarial ensemble that involves a group of adversaries. Figure 1 presents an intuitive demonstration of the advantages of an adversarial ensemble. In this work, we first theoretically establish that an adversary ensemble can relieve the first issue by proving that it can efficiently estimate the solution and the optimal value of the inner minimization problem, alleviating the over-optimism. Next, we employ the proposed adversarial ensemble to mitigate the over-pessimism by altering the objective of the RL agent from the original worst-case performance to *the average performance of the worst-k adversaries* (hence the name structured adversarial ensemble). By addressing these problems, our proposed method significantly outperforms other robust RL baselines, corroborating the importance of these identified problems. Extensive experiments on a wide range of tasks with strong baselines have demonstrated that the policies generated by our method has enhanced robustness, and the improved robustness is consistent across various types of environmental disturbance.

## 2 PRELIMINARY

For any finite set $A$, we use $|A|$ to denote its cardinality. For any positive integer $m$, we use $[m]$ to represent the set of integers $\{1, \ldots, m\}$. For any set $\mathcal{M}$, we use $\Delta(\mathcal{M})$ to denote the set of all possible probability measures over the Borel $\sigma$-algebra of $\mathcal{M}$. In this work, we consider a Markov Decision Process (MDP) with adversaries in the environment, defined by a tuple of 6 elements $(\mathcal{S}, \mathcal{A}^p, \mathcal{A}^a, \mathcal{P}, r, \gamma, p_0)$; here, $\mathcal{S}$ is the set of states, $\mathcal{A}^p/\mathcal{A}^a$ are the sets of actions that the agent (protagonist) or adversaries can take, $\mathcal{P} : \mathcal{S} \times \mathcal{A}^p \times \mathcal{A}^a \to \Delta(\mathcal{S})$ is the transition function that describes the distribution of the next state given the current state and actions taken by the agent and the adversaries, $r : \mathcal{S} \times \mathcal{A}^p \times \mathcal{A}^a \to \mathbb{R}$ is the reward function for the agent (we set the reward function for the adversary to $-r$ as we consider a zero-sum game framework in this work), $\gamma \in [0, 1)$ is the discounting factor, and $p_0$ is the distribution of the initial state. We use $\pi_\theta : \mathcal{S} \to \Delta(\mathcal{A}^p)$ and $\pi_\phi : \mathcal{S} \to \Delta(\mathcal{A}^a)$ to respectively denote the polices of the agent and the adversaries, where $\theta$ and $\phi$ are their parameters. Specifically, we use $\pi_{\phi_i}$ and $\phi_i$ to denote the policy of the $i$-th adversary and its parameter. Let $s_t \in \mathcal{S}$ be the state of the environment at time $t$, $a_t^p \in \mathcal{A}^p$ (respectively $a_t^a \in \mathcal{A}^a$) the action of the agent (respectively adversary) at time $t$. We use

$$R(\theta, \phi) \doteq \mathbb{E}_{s_0 \sim p_0} \Big[ \sum_{t=0}^{\infty} \gamma^t r(s_t, a_t^p, a_t^a) | a_t^p \sim \pi_\theta(s_t), a_t^a \sim \pi_\phi(s_t) \Big] \tag{1}$$

to represent the cumulative discounted reward that the agent $\pi_\theta$ can receive under the disturbance of the adversary $\pi_\phi$. The objective of adversarial training (two-player max-min game) for robustness (Pinto et al., 2017; Vinitsky et al., 2020) is defined as follows:

$$\max_{\theta \in \Theta} \min_{\phi \in \Phi} R(\theta, \phi), \tag{2}$$

where $\Theta$ and $\Phi$ are pre-defined parameter spaces for the agent and the adversaries. In this approach, the RL agent maximizes the worst-case performance under disturbance.

# 3 ROBUSTNESS THROUGH ADVERSARIAL ENSEMBLE

Although adversarial training has achieved great empirical success, two major challenges persist. First, it is challenging to obtain a close approximation of the optimal solution $\phi^* \in \Phi$ to the inner minimization problem in equation 2. This can result in an over-optimistic estimation of the worst case performance of the RL agents. Second, an imprecise choice of the candidate adversary set $\Phi$ will result in an over-conservative agent if it is distracted by unlikely scenarios during learning. To address these challenges, we propose to employ an adversarial ensemble which involves a group of adversaries. In this section, our algorithm will be presented along with the theoretical results that illustrate the motivations and justify its effectiveness. Specifically, in Section 3.1, we first establish that introducing an adversarial ensemble can alleviate the over-optimism by proving that it can help estimate the solutions to the inner optimization problem efficiently, *i.e.*, the required size of an ensemble for a desired approximation precision is amiable. In Section 3.2, we propose a new objective to replace the worst-case performance optimization in equation 2 to prevent the trained agents from being over-conservative. In Section 3.3, we summarize and present detailed steps of the proposed algorithm.

## 3.1 ADVERSARIAL ENSEMBLE

Here we present the motivation for introducing an adversarial ensemble and theoretically establish its advantage over a single adversary. Proofs to all the theoretical results are deferred to Appendix B. Due to the complexity of $R(\theta, \phi)$, the most popular approach to solve the inner optimization problem for a given $\theta$ is to use a single adversary and update the adversary with first-order optimization method such as gradient descent. However, this approach is likely to be stuck in the local optima as $R(\theta, \phi)$ is often highly non-convex over $\phi$, deviating from the global optimal solution and value of the inner problem. To address this issue, we first propose a variation of the above approach that employs multiple adversaries. Specifically, instead of a single adversary that updates itself, we employ a set of *fixed* adversaries denoted by $\widehat{\Phi} \doteq \{\phi_i\}_{i=1}^m$, where $m$ is the total number of adversaries and for all $i \in [m]$, $\phi_i \in \Phi$. Subsequently, we transform the original optimization problem in equation 2 into the following one

$$\max_{\theta \in \Theta} \min_{\phi \in \widehat{\Phi}} R(\theta, \phi); \tag{3}$$

in the new objective the agent $\pi_\theta$ still optimizes the worst-case performance but only over a finite set of adversaries. A direct methodological advantage of this approach over the original one is that there is no need to use a first-order method. To find the optimal solution and value of the inner minimization problem in equation 3, one only need to approximate $R(\theta, \phi_i)$ for all $\phi_i$ in $\widehat{\Phi}$, and to select the adversary $\phi$ that results in the minimum $R(\theta, \phi)$. This process takes linear time with respect to the number of adversaries. Note in this approach, only the 1-dimensional $R(\theta, \phi)$ needs to be approximated. However, in the original approach, to update the adversary, the gradient of $R(\theta, \phi)$ (with respect to $\phi$) must be estimated, which is a $d_\phi$-dimensional object where $d_\phi$ is the dimension of $\phi$ and often a large number. We next prove that the proposed approach can efficiently approximate the inner optimization problem in equation 2.

**Definition 1** ($L^\infty$ Norm). *For a function $h : \mathcal{X} \to \mathbb{R}$, we define its $L^\infty$ norm as $||h||_\infty = \sup_{x \in \mathcal{X}} |h(x)|$.*

**Definition 2** ($\epsilon$-packing). *Let $(\mathcal{U}, d)$ be a metric space where $d : \mathcal{U} \times \mathcal{U} \to \mathbb{R}^+$ is the metric function. Then a finite set $\mathcal{X} \subset \mathcal{U}$ is an $\epsilon$-packing if no two distinct elements in $\mathcal{X}$ are $\epsilon$-close to each other, i.e.,*

$$\inf_{x, x' \in \mathcal{X} : x \neq x'} d(x, x') > \epsilon.$$

Let $R_\Phi$ denote a function class defined as

$$R_\Phi \doteq \{R_\phi \doteq R(\theta, \phi) : \Theta \to \mathbb{R} \,|\, \phi \in \Phi\}.$$

Our first result illustrates that if one chooses a set of adversaries that are different enough, then the number of adversaries needed to approximate the inner optimization problem is in approximately linear order of the desired precision.

**Theorem 1.** *Consider the metric space $(R_\Phi, ||\cdot||_\infty)$ where for any two functions $R_\phi, R_{\phi'} \in R_\Phi$, the distance between them is defined as $d(R_\phi, R_{\phi'}) \doteq ||R_\phi - R_{\phi'}||_\infty$. Assume that $R_\Phi$ has finite radius under this metric, i.e.,*

$$\sup_{\phi, \phi' \in \Phi} d(R_\phi, R_{\phi'}) \leq r_{\max}, \tag{4}$$

*where $r_{\max} < \infty$ is a finite number. Let $\widehat{\Phi} = \{\phi_i\}_{i=1}^m \subset \Phi$. If $R_{\widehat{\Phi}}$ is a maximal $\epsilon$-packing then $|R_{\widehat{\Phi}}| \geq \lceil \frac{r_{\max}}{\epsilon} \rceil$, where $\lceil c \rceil$ is the smallest integer that is larger than or equal to $c$. Moreover, for any $\theta \in \Theta$, let $\widehat{\phi} \doteq \arg\min_{\phi \in \widehat{\Phi}} R(\theta, \phi)$ denote the approximated solution and $\phi^* \doteq \arg\min_{\phi \in \Phi} R(\theta, \phi)$ denote the optimal solution. Then, the approximation error of $\widehat{\phi}$ on the inner optimization problem is upper bounded by $\epsilon$, i.e.,*

$$|R(\theta, \phi^*) - R(\theta, \widehat{\phi})| \leq \epsilon.$$

The assumption in equation 4 is essentially requesting that for any policy $\pi_\theta$, its performance in two different environments cannot vary infinitely. This is a common condition satisfied by any RL problems with finite reward functions. From another perspective, this is equivalent to suggesting that the adversary cannot be omnipotent. Under this assumption, if we can construct a set of adversaries that are distinct from each other, then the number of adversaries one needs for approximation is about $O(\frac{1}{\epsilon})$, where $\epsilon$ can be interpreted as the desired level of accuracy towards the approximation. We next show that if one only wants to use an adversarial ensemble to approximate accurately with high probability, instead of an almost sure approximation as in Theorem 1, then the number of required adversaries can be reduced.

**Theorem 2.** *Assume that $\Phi$ is a metric space with a distance function $d : \Phi \times \Phi \mapsto \mathbb{R}$. Let $\sigma$ be any probability measure on $\Phi$. Let $\widehat{\Phi} = \{\phi_i\}_{i=1}^m$ be a set of independently sampled elements from $\Phi$ following identical measure $\sigma$. Consider a fixed $\theta \in \Theta$ and assume that $R(\theta, \phi)$ is an $L_\phi$-Lipschitz continuous function of $\phi$ with respect to the metric space $(\Phi, d)$. Let $\widehat{\phi}$ and $\phi^*$ be defined the same as in Theorem 1. For presentation simplicity, assume that $\sigma(\{\phi : d(\phi, \phi^*) \leq \epsilon\}) \geq L_\sigma \epsilon$. Let $0 < \delta < 1$ denote the probability of a bad event. Then with probability $1 - \delta$, the approximation error of $\widehat{\phi}$ on the inner optimization problem is upper bounded by $\epsilon$ if $m \geq \log(\delta) \log^{-1}(1 - \frac{L_\sigma}{L_\phi}\epsilon)$.*

In Theorem 2, one can replace $L_\sigma$ with other dense conditions about measure of $\Phi$ and reach similar results. Compared with Theorem 1, if one can sample from a measure that is dense around the optimal $\phi$, then the required number of adversaries can be decreased. Specifically, if one would like to decrease of probability of bad approximation by half, the extra number of adversaries needed is about $O(\frac{1}{c})$ where $c$ is a constant related to how dense one can sample close to the true optimal.

While the above results shed some lights on how we should design the adversarial ensemble algorithm, one may still encounter a couple of challenges in practice. In Theorem 1, we would like to construct an $\epsilon$-packing. However, as even verifying for two adversaries $\phi, \phi'$ that $d(R_\phi, R_{\phi'}) = ||R_\phi - R_{\phi'}||_\infty \geq \epsilon$ is challenging, it makes construction of an $\epsilon$-packing to be intractable. In Theorem 2, it is often challenging to estimate $L_\phi$ as well as to construct a measure $\sigma$ that is dense near $\phi^*$. To address these problems, we let $\phi_i \in \widehat{\Phi}$ be learners, instead of fixed adversaries. The objective then becomes

$$\max_{\theta \in \Theta} \min_{\phi_1, \ldots, \phi_m \in \Phi} \min_{\phi \in \{\phi_i\}_{i=1}^m} R(\theta, \phi). \tag{5}$$

It is important and interesting to observe that the solution set of equation 5 is identical to that of the maximin problem in the original approach.

**Lemma 3.** *The solution set to the optimization problem in equation 2 is identical to the solution set of the optimization problem in equation 5. That is, for any $\theta \in \Theta$ and integer $m \geq 1$,*

$$\min_{\phi \in \Phi} R(\theta, \phi) = \min_{\phi_1, \ldots, \phi_m \in \Phi} \min_{\phi \in \{\phi_i\}_{i=1}^m} R(\theta, \phi).$$

**Insights from the Theoretical Results.** From an intuitive perspective, Theorem 1 and Theorem 2 reveal that when the adversaries in the ensemble are distinct to each other, the accuracy for approximating the true worst-case performance can be efficiently improved with increased number of adversaries. Lemma 3 implies that the true benefit brought by the adversarial ensemble lies in the optimization process instead of the final optimal solution it offers. In other words, adversarial training with an ensemble of adversaries still optimizes the worst-case performance of an agent over a pre-defined candidate adversary set $\Phi$, but adversarial ensemble can alleviate the challenge brought by the inner optimization. Importantly, these results assure us that the required size of the adversarial ensemble to improve performance is not overwhelming. To verify the correctness of these insights, we conduct empirical study about the effect of the number of adversaries on the performance of the RL agents (see Section 4), and found that even with only 10 adversaries the robustness of agents can still be significantly improved.

## 3.2 RESOLVING POTENTIAL OVER-PESSIMISM

The max-min game in equation 2 can lead to a solution that is too conservative due to the worst case optimization if the range of the adversaries $\Phi$ is not chosen correctly. Specifically, as the max-min problems in robust RL are normally solved by iterative updates of the protagonist and the adversaries, where in each iteration we have an adversary $\phi$ against whom we will optimize the protagonist. However, if the adversary set is not precise, $\phi$ may be a mis-specified scenario. If the rest $k-1$ adversaries (or the majority of the worst-$k$ adversaries) are indeed in the true interested scenarios, optimizing the average over the worst-k adversaries distracts the attention of the protagonist from the single uninterested worst case to the cases of interest.

To this end, we modify the objective of the agent $\pi_\theta$, from optimizing its worst-case performance to optimizing its average performance over the worst-$k$ adversaries. We define the worst-$k$ adversaries in a set of adversaries $\{\phi_i\}_{i=1}^m$ for a fixed agent $\pi_\theta$ as follows. A group of $k$ adversaries is the worst-$k$ adversaries if the expected cumulative rewards received by the agent $\pi_\theta$ under their attack are smaller than that under the attack from the rest $m-k$ adversaries. Specifically, for a given set of adversaries $\widehat{\Phi} \doteq \{\phi_i\}_{i=1}^m$ and $\theta$, let $W_\theta(\phi) \doteq \{\phi' \in \widehat{\Phi} : R(\theta, \phi') \leq R(\theta, \phi)\}$. For an integer $k \geq 1$, let $I_{\theta,\widehat{\Phi},k} \doteq \{i \in [m] : \phi_i \in \widehat{\Phi}, |W_\theta(\phi_i)| \leq k\}$ denote the set of indices of the worst-$k$ adversaries for a given policy $\pi_\theta$. The new objective is then defined as:

$$\max_{\theta \in \Theta} \min_{\phi_1,\ldots,\phi_m \in \Phi} \frac{1}{|I_{\theta,\widehat{\Phi},k}|} \sum_{i \in I_{\theta,\widehat{\Phi},k}} R(\theta, \phi_i). \tag{6}$$

Average over worst-$k$ performances can balance out the pessimism, preventing the agent from attaching to the scenarios that can potentially lead to over-conservative policies.

## 3.3 ROBUST REINFORCEMENT LEARNING WITH STRUCTURED ADVERSARIAL ENSEMBLE

We now introduce our algorithm, _Robust Reinforcement Learning with Structured Adversarial Ensemble_ (ROSE) in Algorithm 1. ROSE is an iterative algorithm that sequentially update the policy $\pi_\theta$ and the adversarial ensemble $\{\phi_i\}_{i=1}^m$ to solve

$$\max_{\theta \in \Theta} \min_{\phi_1,\ldots,\phi_m \in \Phi} \frac{1}{|I_{\theta,\widehat{\Phi},k}|} \sum_{i \in I_{\theta,\widehat{\Phi},k}} R(\theta, \phi_i),$$

where $R(\theta, \phi) = \mathbb{E}\left[\sum_{t=0}^\infty \gamma^t r_t | \pi_\theta, \pi_\phi\right]$ is the expected (discounted) cumulative rewards that the agent $\pi_\theta$ can receive under the disturbance of the adversary $\pi_\phi$. For ease of presentation, we assume that all the rollout trajectories have length $H$. We will use superscript to denote the index of iteration number. For instance, $\phi_i^t$ denotes the parameter of the $i$-th adversary in the $t$-th iteration of the algorithm. ROSE first randomly initialize the agent policy and the adversarial ensemble. In each iteration, we first update the adversary ensemble and then update the agent policy with the updated adversaries. Specifically, in the $t$-th iteration, for $i \in [m]$, we collect a batch of trajectories $\rho_i^t = \{\tau_i^{t,j}\}_{j=1}^{b_a}$ where $b_a$ is the batch size for training the adversarial ensemble. The trajectories are collected by rolling out the agent $\pi_\theta$ and the $i$-adversary in the environment. Each trajectory in $\rho_i^t$ consists of $H$ transition tuples $\{(s_0, a_0, -r_0, s_1) \times \cdots \times (s_H, a_H, -r_H, s_{H+1})\}$, where for

$0 \leq h \leq H$, $a_h$ is the action by the $i$-th adversary and $r_h$ is the reward received by the agent. After collecting the trajectories for all the adversaries, we use these trajectories to estimate $R(\theta, \phi_i)$ for all $i \in [m]$, and select the worst-$k$ adversaries. Then we update these $k$ selected adversaries with the corresponding trajectories. The rest $m - k$ adversaries remain unchanged. Note that any RL algorithms can be used in the update. After the adversarial ensemble has been updated, we update the agent policy $\pi_\theta$. To identify the worst-$k$ adversaries, i.e., the elements in $I_{\theta, \widehat{\Phi}, k}$, we first estimate $R(\theta, \phi_i)$ for $i \in [m]$ by rolling out the agent $\pi_\theta$ with the $i$-th adversary in the environment to have an estimation $\widehat{R}_i$. Then we set $I_{\theta, \widehat{\Phi}, k}$ to contain all the indices $i$ such that $\widehat{R}_i$ is no greater than the $k$-th smallest element of the set $\{\widehat{R}_j\}_{j=1}^m$. For each adversary $i$ in $I_{\theta, \widehat{\Phi}, k}$, we roll out the agent $\pi_\theta$ with $\pi_{\phi_i}$ to collect $b_p$ trajectories, each trajectory consisting of $\{(s_0, a_0, r_0, s_1) \times \cdots \times (s_H, a_H, r_H, s_{H+1})\}$, where for $0 \leq h \leq H$, $a_h$ is the action by the agent $\pi_\theta$ and $r_h$ is the reward received by the agent. Then we pull all the collected trajectories together as the training dataset $\rho_p^t$ with $k \cdot b_p$ trajectories in total. Finally we use Trust Region Policy Optimization (TRPO) (Schulman et al., 2015)[1] to update $\theta$, i.e., the parameter of the agent, with $\rho_p^t$. The proposed algorithm is executed until the parameter of the agent policy $\theta$ converges or for a maximum of $T$ iteration, whichever happens first.

## 4 EXPERIMENTS

In this section, we empirically evaluate ROSE with the following baselines: (*i*) RL agents trained without adversarial training, (*ii*) RARL (Robust Adversarial Reinforcement Learning): RL agent trained against a single adversary in a zero-sum game (Pinto et al., 2017), (*iii*) RAP (Robustness via Adversary Populations): agent trained with a uniform sampling from a population of adversaries (Vinitsky et al., 2020), and (*iv*) M2TD3 (Tanabe et al., 2022): a state-of-the-art (SOTA) method for robust RL which, in contrast to all the baselines with adversarial training, requires information about the uncertainty set of the environment. We note that, despite the additional information, ROSE still outperforms M2TD3 in most scenarios with adversarial attacks (see Table 1), further corroborating the importance of the identified problems and the value of ROSE.

We investigate 2 types of robustness: (*a*) robustness to disturbance on the agent (e.g., action noise and adversarial policies) and (*b*) robustness to environmental change (e.g., mass and friction). For fairness and consistency of the performance, we use TRPO to update policies for all baselines as well as ROSE. Our adversarial setting follows Pinto et al. (2017), where the adversary learns to destabilize the protagonist by applying forces on specific points, which is denoted by red arrows in Figure 8. The details of the experiments can be found in Appendix E.

Table 1: Performance of ROSE and baselines under various disturbances using TRPO.

| Method | Baseline (0 adv) | RARL (1 adv) | RAP (population adv) | ROSE (ours) | M2TD3 |
|---|---|---|---|---|---|
| Ant (No disturbance) | 0.77±0.16 | 0.81±0.12 | 0.83±0.08 | **0.87±0.13** | 0.84±0.22 |
| Ant (Action noise) | 0.66±0.19 | 0.67±0.16 | 0.67±0.09 | **0.70±0.14** | 0.66±0.16 |
| Ant (Worst Adversary) | 0.21±0.18 | 0.25±0.17 | 0.30±0.14 | **0.38±0.16** | 0.29±0.11 |
| InvertedPendulum (No disturbance) | **1.00±0** | 0.96±0.11 | 0.99±0.04 | 0.99±0.03 | **1.00±0** |
| InvertedPendulum (Action noise) | 0.91±0.13 | 0.91±0.15 | 0.95±0.10 | 0.96±0.13 | **0.97±0.16** |
| InvertedPendulum (Worst Adversary) | 0.86±0.16 | 0.88±0.18 | 0.90±0.19 | **0.92±0.12** | 0.90±0.21 |
| Hopper (No disturbance) | 0.78±0.003 | 0.79±0.02 | 0.84±0 | 0.95±0.01 | **0.97±0.11** |
| Hopper(Action noise) | 0.71±0.001 | 0.74±0.004 | 0.80±0 | **0.91±0.006** | 0.77±0.07 |
| Hopper (Worst Adversary) | 0.42±0.03 | 0.54±0.04 | 0.70±0.007 | **0.84±0.14** | 0.83±0.25 |
| Half-Cheetah (No disturbance) | 0.77±0.05 | 0.72±0.03 | 0.76±0.02 | **0.87±0.05** | 0.81±0.06 |
| Half-Cheetah(Action noise) | 0.59±0.2 | **0.76±0.04** | 0.67±0.1 | **0.76±0.16** | 0.68±0.13 |
| Half-Cheetah (Worst Adversary) | 0.16±0.1 | 0.19±0.05 | 0.24±0.36 | **0.52±0.21** | 0.50±0.10 |
| Walker2d (No disturbance) | 0.85±0.27 | 0.84±0.43 | 0.43±0.02 | 0.84±0.44 | **0.88±0.31** |
| Walker2d (Action noise) | 0.78±0.31 | 0.80±0.28 | 0.36±0.04 | **0.83±0.37** | 0.79±0.21 |
| Walker2d (Worst Adversary) | 0.36±0.26 | 0.34±0.12 | 0.34±0.22 | **0.68±0.23** | 0.21±0.43 |

**Robustness to Agent Disturbance.** To investigate robustness to action disturbance, we conduct experiments on the Ant, InvertedPendulum, Hopper, Half-Cheetah, and Walker2d continuous control

---

[1]This can be generalized to any RL policy optimization method. We provide ablation studies in Section 4 to investigate the effect of the RL algorithm that implements ROSE.

tasks in MuJoCo environments. To measure robustness to such effect, we report the normalized return of the learned policies in Table 1 for 3 types of disturbances during evaluation: (*i*) no disturbance, (*ii*) random adversary that adds noise to the actions of the agents, and (*iii*) the worst adversary that represents the worst case performance of a given policy. To provide such an extreme disturbance in (*iii*), for each policy trained either by a baseline method or ROSE, we train an adversary to minimize its reward while holding the parameters of that policy as constant, and this process is repeated with 10 random seeds. In other words, the trained policies undergo disturbances from distinct adversaries, specifically trained to minimize their rewards. In Table 1, we first show that learning with adversaries improves the performance compared with the baseline ($1^{st}$ column in Table 1) even though there is no change between training and testing conditions for the baseline, an observation also reported by Pinto et al. (2017). We also emphasize that ROSE outperforms RAP under disturbance, which supports our argument that simply averaging over all the adversaries may decrease robustness. We observe that M2TD3 training with an uncertainty parameter set is relatively competitive in the environment without disturbance while our ROSE demonstrates its strength in robustness to the action noise and learned adversarial policy.

**Robustness to Test Conditions (Environmental Change).**   In addition to being robust to external disturbance, robustness should also be reflected in different internal conditions. We consider robustness to the conditions of the test environments, such as mass and friction, which are critical parameters for locomotion tasks in the MuJoCo environment. We conduct experiments on the Ant, InvertedPendulum, Hopper, Half-Cheetah, and Walker2d continuous control tasks in MuJoCo environments. During training, all the policies across different methods are trained in the environment with a specific pair of mass and friction values. To evaluate the robustness and generalization of the learned policies, we test the policies in distinct environments with jointly varying mass and friction coefficients. As shown in Figure 2, our method (ROSE) has competitive performance (significantly improved performance in Hopper and Half-Cheetah) under varying test conditions. Notably, ROSE has demonstrated symmetric robustness with respect to varying mass and friction in Hopper task ($1^{st}$ row (a)-(d) in Figure 2) where we set both the friction and mass coefficients equal to $1.0$ during training. It can observed that the performance of ROSE is symmetric under decrease/increase of the coefficients centered at $1.0$, the training coefficients. The performance of RAP and other baselines does not demonstrate this trend. Moreover, when tested in environments that gradually shift away from the training environments, the performance drop of ROSE is less rapid compared to other baselines. This demonstrates the stability and predictability of ROSE. In Figure 7 in Appendix, We provide additional experimental results about the distribution of the rewards of various methods in distinct environments. Compared with other baselines, the rewards of ROSE are more centered in the high-reward region and there is no extremely low rewards, further demonstrating the efficacy of our approach. Note we omit evaluation of M2TD3 with varying test conditions since M2TD3 is already trained with additional information on mass and friction values.

**Ablation Studies.**   Here we provide ablation studies to better understand: (A1) the gain of addressing the potential over-pessimism; (A2) the effect of the total number of adversaries; (A3) the effect of value of $k$ in the worst-$k$ set; (A4) the update frequencies of all adversaries in ROSE; (A5) the effect of the underlying RL algorithm that implements ROSE.

**A1.** To understand the benefits of addressing over-pessimism, we investigate a variation of ROSE (referred to as *ROSE-all*) where instead of updating the worst-$k$ adversaries we update all the adversarial policies. To validate our analysis in Section 3, we conduct experiments in Hopper environment and cross-validate the robustness. Empirical evidence demonstrates that ROSE significantly outperforms ROSE-all. Due to space limitation, please refer to Appendix C.1 and C.2 for details.

**A2/A3.** We vary the size the adversarial ensemble and the value of $k$ in the worst-$k$ set. As can be seen from Table 2, when the value of $k$ increases, we are approaching RAP and focusing less on worst-case optimization. When the value of $k$ decreases too much, the performance also decreases. This aligns with our conjecture that a single adversary can get trapped into extreme cases, also leading to degraded performance.

**A4.** It is theoretically possible that the worst adversaries stay worst and thus untrained. However, we find in practice if initialized differently, the worst-k adversaries keep changing and all adversaries are updated frequently. We conduct an experiment to verify this, and result is deferred to the Appendix

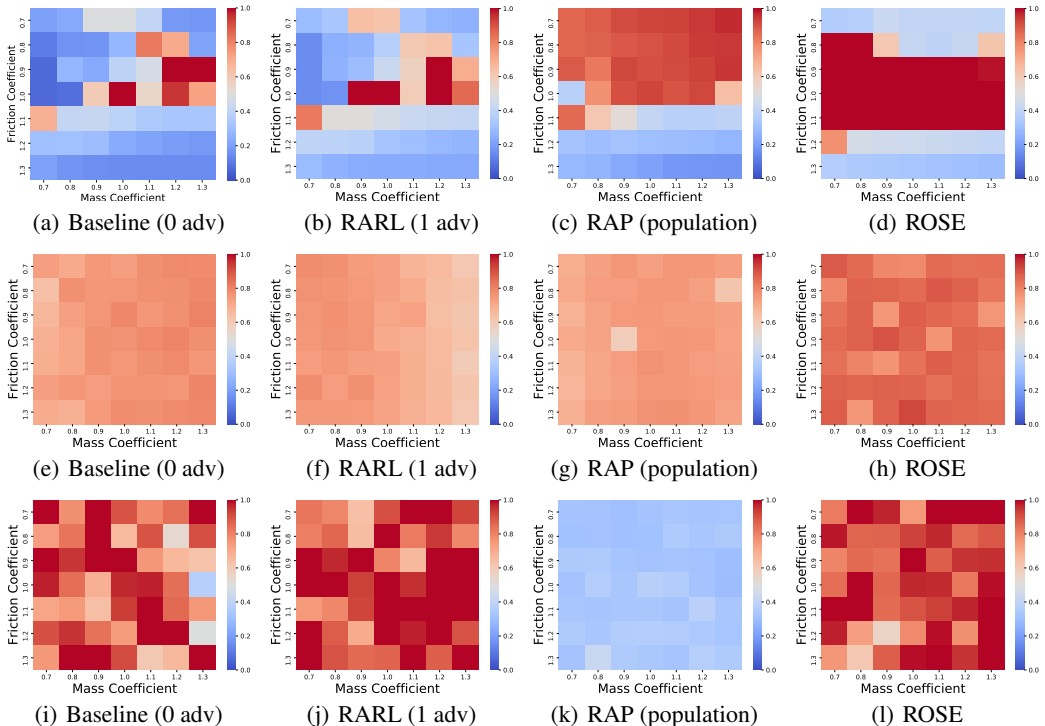

Figure 2: Average normalized return across 10 seeds tested via different mass coefficients on the x-axis and friction coefficients on the y-axis. High reward has red color; low reward has blue color. $1^{st}$ row: Hopper, $2^{nd}$ row: Half-Cheetah, $3^{rd}$: Walker2d

due to space constraints. As can be seen in Figure 6 in the Appendix, the updates are distributed evenly across adversaries, demonstrating that the worst-k adversaries keep changing.

**A5.** To ensure that the superior performance of ROSE is consistent, we conduct additional experiments where all the baselines and ROSE are implemented with Proximal Policy Optimization (PPO) (Schulman et al., 2017b) and Deep Deterministic Policy Gradient (DDPG) (Lillicrap et al., 2019). Notably, DDPG is an off-policy RL algorithm in contrast to the on-policy TRPO and PPO. Due to space constraints, please see Appendix D for details. It can be observed that ROSE maintains its strong performance across various of RL algorithms for implementation.

Table 2: Ablation Studies of Number of $k$ in Half-Cheetah Environment.

| Number of Adversaries $N$ | 5 | | | 10 | | | 20 | | |
|---|---|---|---|---|---|---|---|---|---|
| worst $k$ with percentage of $N$ | 10% | 30% | 50% | 10% | 30% | 50% | 10% | 30% | 50% |
| No disturbance | 0.73±0.02 | 0.73±0.04 | 0.70±0.03 | 0.74±0.05 | 0.87±0.05 | 0.84±0.04 | 0.73±0.03 | 0.81±0.04 | 0.78±0.06 |
| Action Noise | 0.63±0.22 | 0.68±0.23 | 0.61±0.18 | 0.65±0.18 | 0.76±0.16 | 0.74±0.15 | 0.60±0.17 | 0.73±0.21 | 0.72±0.23 |
| Worst Adversary | 0.23±0.11 | 0.21±0.15 | 0.18±0.09 | 0.36±0.15 | 0.52±0.21 | 0.43±0.26 | 0.33±0.19 | 0.44±0.18 | 0.40±0.24 |

## 5 RELATED WORKS

Recent deep RL advancements, over TD learning (Kostrikov et al., 2021; Kumar et al., 2020), actor-critic (Haarnoja et al., 2018; Lee et al., 2020), model-based (Hafner et al., 2019; Kaiser et al., 2019) and RvS (Chen et al., 2021; Emmons et al., 2021) methods, have significantly impacted how autonomous agents can facilitate efficient decision making in real-world applications, including healthcare (Gao et al., 2022; Tang & Wiens, 2021), robotics (Ibarz et al., 2021; Kalashnikov et al., 2018), natural language processing (Ziegler et al., 2019), etc. However, the large parameter search

space and sample efficiency leave the robustness of RL policies unjustified. Consequentially, there exists a long line of research investigating robust RL (Moos et al., 2022).

One research topic closely related to our method is domain randomization, which is a technique to increase the generalization capability over a set of pre-defined environments. The set of environments are parameterized (e.g., friction and mass coefficient) to allow the agent to encode the knowledge about the deviations between training and testing scenarios. The environment parameters are commonly uniformly sampled during training (Tobin et al., 2017; Peng et al., 2018; Siekmann et al., 2021; Li et al., 2021b). Even though ADR (Mehta et al., 2020) is proposed to learn a parameter sampling strategy on top of domain randomization, all of the aforementioned methods are not learned over the worst-case scenarios. Moreover, in real-life applications, if not chosen carefully, the environment set can also lead to over-pessimism with a larger range while selecting a smaller range of the set will be over-optimistic. Hence, our proposed method can be readily extended into domain randomization by considering the environments as fixed adversaries.

Robustness to transition models has been widely investigated. It was initially studied by robust MDPs (Bagnell et al., 2001; Iyengar, 2005; Nilim & Ghaoui, 2003) through a model-based manner by assuming the uncertainty set of environmental transitions is known, which can be solved by dynamic programming. In this approach, a base dynamic model is assumed and the uncertainty set is crafted as a ball centered around the base model with a predetermined statistical distance or divergence, *e.g.,* KL-divergence or Wasserstein distance. Following works address scenarios where the base model is unknown but samples from the base model are available. For example, Panaganti & Kalathil (2022); Shi et al. (2023) propose model-based algorithms that first estimates the base model and then solve the robust MDP; Panaganti & Kalathil (2021); Roy et al. (2017) propose online model-free policy evaluation and policy iteration algorithms for robust RL with convergence guarantees; Xu et al. (2023) proposes algorithms with polynomial guarantees for tabular cases where both the number of states and actions are finite.; Panaganti et al. (2022); Shi & Chi (2022) further extends the study of robust RL with only offline data. In contrast to these works, we follow the approach of RARL which does not explicitly specify the set of environments but learns a robust policy by competing with an adversary. Subsequent works generalize the objective to unknown uncertainty sets, and formulate the uncertainty as perturbations/disturbance introduced into the environments (Shi et al., 2023; Abraham et al., 2020; Tanabe et al., 2022; Vinitsky et al., 2020; Pinto et al., 2017). Notably, RARL (Pinto et al., 2017) introduces an adversary with the objective to affect the environment to minimize the agent's rewards. Notably, while in this work we focus on robustness to the transition model, there are two other types of robustness: robustness to the disturbance of actions (Tessler et al., 2019; Li et al., 2021a) and robustness to state/observation (Zhang et al., 2021; He et al., 2023). There are meta-RL works that tackle distributional shift across tasks (Lin et al., 2020; Zahavy et al., 2021), which are orthogonal to the type of robustness we consider. We also distinguish the difference in set-ups between our work and several works. Specifically, Shen & How (2021) focuses on the scenario where there are other agents with unknown objectives and employs an ensemble to simulate the behaviors of these agents but not for a policy with robustness to environmental disturbance; Huang (2022) employs Stackelberg game to address the potential over-conservatism in scenarios where the adversaries do not act simultaneously, while our work follows the conventional Nash equilibrium widely employed by the robust RL works (Moos, 2022); Zhai (2022) proposes to adaptively scale the weights of of a set of adversaries to improve stability and robustness, while our method employs the ensemble differently to address the over-pessimism caused by potential misspecification of the adversary set. Moreover, our work additionally establishes theoretical support and rigorous understanding for the application of ensemble methods in robust RL, which is the element missing in these works.

# 6 DISCUSSION

We have proposed a new algorithm ROSE that employs an adversarial ensemble to address two important challenges in adversarial training for robust RL: the over-optimism and over-pessimism. Experimental results on diverse RL environments corroborate that ROSE can generate policies robust to a variety of environmental disturbance. One limitation of our work is the extra computation power required by the adversarial ensemble. However, our algorithm can be easily distributed and paralleled as the adversaries attack independently. Another interesting problem worth investigation is the convergence conditions of the RL agents under adversarial training. We will pursue this question in our future work.

## REPRODUCIBILITY STATEMENT

We have submitted the code of our implementation of ROSE as supplementary material. Information about the benchmarks are detailed in Section 4. The experimental details including the values of hyper-parameters are elaborated in Section 4 and in Appendix E.

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

APPENDIX

## A    ALGORITHM

---

**Algorithm 1** Robust Reinforcement Learning with Structured Adversarial Ensemble (ROSE)

---

**Input:** $m$: size of the adversarial ensemble ; $k$: the number of the worst adversaries to use; $\lambda_p$: step size for updating the agent policy; $\lambda_a$: step size for updating the adversary ensemble;

**Output:** $\widehat{\theta}$: parameter for the agent policy.

  Randomly initialize $\theta$ and $\{\phi_i\}_{i=1}^m$

  $t \leftarrow 0, \theta^t \leftarrow \theta, \phi_i^t \leftarrow \phi_i \ \ \forall i \in [m]$

  **for** $t = 0 : T - 1$ **do**

    {Update the adversarial ensemble.}

    **for** $i = 1 : m$ **do**

      Estimate $R(\theta^t, \phi_i^t)$ by rolling out the agent $\pi_{\theta^t}$ with the adversary $\pi_{\phi_i^t}$

    **end for**

    Construct $I_{\theta, \widehat{\Phi}, k}$ with the estimations.

    $\phi_j^{t+1} \leftarrow \phi_j^t - \lambda_a \nabla_\phi R(\theta^t, \phi_j^t) \quad \forall j \in I_{\theta, \widehat{\Phi}, k}$

    {Update the agent policy.}

    **for** $i = 1 : m$ **do**

      Estimate $R(\theta^t, \phi_i^{t+1})$ by rolling out the agent $\pi_{\theta^t}$ with the adversary $\pi_{\phi_i^{t+1}}$

    **end for**

    Construct $I_{\theta, \widehat{\Phi}, k}$ with the estimations.

    $\theta^{t+1} \leftarrow \theta^t - \lambda_p \sum_{j \in I_{\theta, \widehat{\Phi}, k}} \nabla_\theta R(\theta^t, \phi_j^{t+1})$

  **end for**

  $\widehat{\theta} \leftarrow \theta^T$

---

## B    PROOFS OF THEORETICAL RESULTS

**Theorem 1.** *Consider the metric space* $(R_\Phi, || \cdot ||_\infty)$ *where for any two functions* $R_\phi, R_{\phi'} \in R_\Phi$, *the distance between them is defined as*

$$d(R_\phi, R_{\phi'}) \doteq ||R_\phi - R_{\phi'}||_\infty.$$

*Assume that* $R_\Phi$ *has finite radius under this metric, i.e.,*

$$\sup_{\phi, \phi' \in \Phi} d(R_\phi, R_{\phi'}) \le r_{\max}, \tag{7}$$

*where* $r_{\max} < \infty$ *is a finite number. Let* $\widehat{\Phi} = \{\phi_i\}_{i=1}^m \subset \Phi$. *If* $R_{\widehat{\Phi}}$ *is a **maximal** $\epsilon$-packing then* $|R_{\widehat{\Phi}}| \ge \lceil \frac{r_{\max}}{\epsilon} \rceil$, *where* $\lceil c \rceil$ *is the smallest integer that is larger than or equal to c, and* $R_{\widehat{\Phi}}$ *is also an* $\epsilon$-net. *Moreover, for any* $\theta \in \Theta$, *let* $\hat{\phi} \doteq \arg\min_{\phi \in \widehat{\Phi}} R(\theta, \phi)$ *denote the approximated solution and* $\phi^* \doteq \arg\min_{\phi \in \Phi} R(\theta, \phi)$ *denote the optimal solution. Then, the approximation error of* $\hat{\phi}$ *on the inner optimization problem is upper bounded by* $\epsilon$, *i.e.,*

$$|R(\theta, \phi^*) - R(\theta, \hat{\phi})| \le \epsilon.$$

*Proof.* Since $R_{\widehat{\Phi}}$ is an $\epsilon$-packing, balls of radius $\frac{\epsilon}{2}$ do not overlap. Consider $\mathcal{U}$ the union of the balls. Any point in $\mathcal{U}$ is clearly within distance $\frac{\epsilon}{2} < \epsilon$ from $R_{\widehat{\Phi}}$. Consider a point $\phi_* \notin \mathcal{U}$. If the ball of radius $\frac{\epsilon}{2}$ around $\phi_*$ is disjoint from $\mathcal{U}$, then $R_{\widehat{\Phi}} \cup \phi_*$ is an $\epsilon$ packing that strictly contains $R_{\widehat{\Phi}}$. This violates the maximality assumption on $R_{\widehat{\Phi}}$. Since $R_{\widehat{\Phi}}$ is an $\epsilon$-packing, then balls of radius $\frac{\epsilon}{2}$ do not overlap. Consider $\mathcal{U}$ the union of the balls. Any point in $\mathcal{U}$ is clearly within distance $\frac{\epsilon}{2} < \epsilon$ from $R_{\widehat{\Phi}}$.

Now, consider a point $\phi_* \notin \mathcal{U}$. If the ball $B(\phi_*, \frac{\epsilon}{2})$ of radius $\frac{\epsilon}{2}$ around $\phi_*$ is disjoint from $\mathcal{U}$ then $R_{\widehat{\Phi}} \cup \phi_*$ is an $\epsilon$-packing that strictly contains $R_{\widehat{\Phi}}$. This violates the maximality of $R_{\widehat{\Phi}}$. Thus $B(\phi_*, \frac{\epsilon}{2})$ has an intersection with at least a ball of radius $\frac{\epsilon}{2}$ around some point of $R_{\widehat{\Phi}}$. It follows from triangle inequality that $\phi_*$ is within distance $\epsilon$ of this point. Since $\phi_*$ was arbitrary, then $R_{\widehat{\Phi}}$ is an $\epsilon$-covering

and an $\epsilon$-net. The fact that $|R_{\widehat{\Phi}}| \geq \lceil \frac{r_{\max}}{\epsilon} \rceil$ follows trivially from the fact that balls of radius $\frac{\epsilon}{2}$ around the points of $R_{\widehat{\Phi}}$ do not intersect and the triangle inequality.

Since $R_{\widehat{\Phi}}$ is an $\epsilon$-net of $R_\Phi$, for any $\phi^*$ there exists $\phi \in \widehat{\Phi}$ such that $||R_\phi - R_{\phi^*}||_\infty \leq \epsilon$. By definition of the $L_\infty$ norm, this implies that for any $\theta \in \Theta$, $|R(\theta, \phi^*) - R(\theta, \phi)| \leq \epsilon$. Also because $\hat{\phi} \doteq \arg\min_{\phi \in \widehat{\Phi}} R(\theta, \phi)$, we have $R(\theta, \hat{\phi}) \leq R(\theta, \phi)$. Since $\phi^*$ is defined as $\arg\min_{\phi \in \Phi} R(\theta, \phi)$, it holds that

$$
\begin{aligned}
|R(\theta, \phi^*) - R(\theta, \hat{\phi})| &= R(\theta, \hat{\phi}) - R(\theta, \phi^*) \\
&\leq R(\theta, \phi) - R(\theta, \phi^*) \\
&= |R(\theta, \phi^*) - R(\theta, \phi)| \leq \epsilon,
\end{aligned}
$$

completing the proof. $\qquad\square$

**Theorem 2.** *Assume that $\Phi$ is a metric space with a distance function $d : \Phi \times \Phi \mapsto \mathbb{R}$. Let $\sigma$ be any probability measure on $\Phi$. Let $\widehat{\Phi} = \{\phi_i\}_{i=1}^m$ be a set of independently sampled elements from $\Phi$ following identical measure $\sigma$. Consider a fixed $\theta \in \Theta$ and assume that $R(\theta, \phi)$ is an $L_\phi$-Lipschitz continuous function of $\phi$ with respect to the metric space $(\Phi, d)$. Let $\hat{\phi}$ and $\phi^*$ be defined the same as in Theorem 1. For presentation simplicity, assume that $\sigma(\{\phi : d(\phi, \phi^*) \leq \epsilon\}) \geq L_\sigma \epsilon$. Let $0 < \delta < 1$ denote the probability of a bad event. Then with probability $1 - \delta$, the approximation error of $\hat{\phi}$ on the inner optimization problem is upper bounded by $\epsilon$ if $m \geq \log(\delta) \log^{-1}(1 - \frac{L_\sigma}{L_\phi}\epsilon)$.*

*Proof.* Assume that we have $\widehat{\Phi} = \{\phi_i\}_{i=1}^m$ as a batch of independently sampled elements from $\Phi$, all following the measure of $\sigma$ during sampling. For any $c > 0$, we have that

$$
\begin{aligned}
&\mathbb{P}(\exists \phi \in \widehat{\Phi} \quad s.t. \quad d(\phi, \phi^*) \leq c) \\
&= 1 - \mathbb{P}(\forall \phi \in \widehat{\Phi} : d(\phi, \phi^*) > c) \\
&= 1 - \mathbb{P}^m(\phi : d(\phi, \phi^*) > c) \\
&= 1 - (1 - \sigma(\{\phi : d(\phi, \phi^*) \leq c\}))^m.
\end{aligned}
\tag{8}
$$

On the other hand, if there exists $\phi \in \widehat{\Phi}$ such that $d(\phi, \phi^*) \leq c$, then by the assumption that $R_\phi$ is $L_\phi$-Lipschitz continuous, $|R(\theta, \phi) - R(\theta, \phi^*)| \leq L_\phi \cdot c$. By definition of $\widehat{\Phi}$, it holds that

$$
\begin{aligned}
|R(\theta, \hat{\phi}) - R(\theta, \phi^*)| &= R(\theta, \hat{\phi}) - R(\theta, \phi^*) \\
&\leq R(\theta, \phi) - R(\theta, \phi^*) = |R(\theta, \phi) - R(\theta, \phi^*)| \\
&\leq L_\phi \cdot c.
\end{aligned}
$$

To prove the theorem, let $c = \frac{\epsilon}{L_\phi}$, and we want

$$
\begin{aligned}
1 - \delta &\leq \mathbb{P}(\exists \phi \in \widehat{\Phi} \quad s.t. \quad d(\phi, \phi^*) \leq c) \\
1 - \delta &\leq 1 - (1 - \sigma(\{\phi : d(\phi, \phi^*) \leq c\}))^m \\
(1 - \sigma(\{\phi : d(\phi, \phi^*) \leq c\}))^m &\leq \delta \\
m &\leq \frac{\log(\delta)}{\log(1 - \sigma(\{\phi : d(\phi, \phi^*) \leq c\})} \\
m &\leq \frac{\log(\delta)}{\log(1 - \frac{L_\sigma}{L_\phi}\epsilon)} \\
m &\leq \log(\delta) \log^{-1}(1 - \frac{L_\sigma}{L_\phi}\epsilon)
\end{aligned}
\tag{9}\tag{10}
$$

where in Eq. equation 9 we use Eq. equation 8 and in equation 10 we use the fact that $c = \frac{\epsilon}{L_\phi}$ and the density assumption that $\sigma(\{\phi : d(\phi, \phi^*) \leq \epsilon\}) \geq L_\sigma \epsilon$. This concludes the proof. $\qquad\square$

**Lemma 3.** *The solution set to the optimization problem in equation 2 is identical to the solution set of the optimization problem in equation 5. That is, for any $\theta \in \Theta$ and integer $m \geq 1$,*

$$
\min_{\phi \in \Phi} R(\theta, \phi) = \min_{\phi_1, \ldots, \phi_m \in \Phi} \min_{\phi \in \{\phi_i\}_{i=1}^m} R(\theta, \phi).
$$

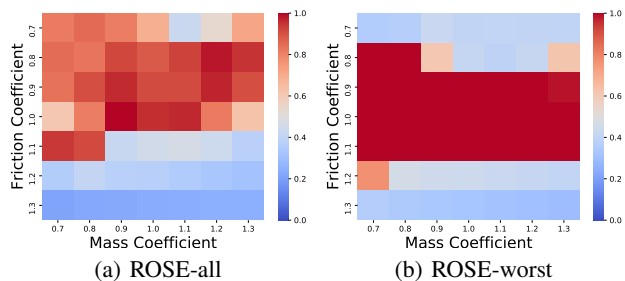

(a) ROSE-all          (b) ROSE-worst

Figure 3: Average normalized return across 10 seeds tested via different mass coefficients on the x-axis and friction coefficients on the y-axis for two variations of ROSE in the Hopper environment. High reward has red color; low reward has blue color.

*Proof.* There are only 3 possibilities regarding the order of $\min_{\phi \in \Phi} R(\theta, \phi)$ and $\min_{\phi_1, \ldots, \phi_m \in \Phi} \min_{\phi \in \{\phi_i\}_{i=1}^m} R(\theta, \phi)$:

- *(i)* $\min_{\phi \in \Phi} R(\theta, \phi) = \min_{\phi_1, \ldots, \phi_m \in \Phi} \min_{\phi \in \{\phi_i\}_{i=1}^m} R(\theta, \phi)$;

- *(ii)* $\min_{\phi \in \Phi} R(\theta, \phi) > \min_{\phi_1, \ldots, \phi_m \in \Phi} \min_{\phi \in \{\phi_i\}_{i=1}^m} R(\theta, \phi)$;

- *(iii)* $\min_{\phi \in \Phi} R(\theta, \phi) < \min_{\phi_1, \ldots, \phi_m \in \Phi} \min_{\phi \in \{\phi_i\}_{i=1}^m} R(\theta, \phi)$.

We prove by contradiction that *(ii)* and *(iii)* are impossible to happen.

If *(ii)* holds, let $\widehat{\Phi}^*$ denote the optimal solution to the right hand side (RHS) and let $\hat{\phi} \doteq \min_{\phi \in \widehat{\Phi}^*} R(\theta, \phi)$. Because *(ii)* holds, we have that $\min_{\phi \in \Phi} R(\theta, \phi) > R(\theta, \hat{\phi})$. However, this is impossible because $\hat{\phi} \in \Phi$ by definition of $\widehat{\Phi}$.

If *(iii)* holds, let $\phi^* \doteq \min_{\phi \in \Phi} R(\theta, \phi)$ be the optimal solution of the left hand side (LHS). Consider any $\widehat{\Phi}$ that includes $\phi^*$, then $\min_{\phi \in \Phi} R(\theta, \phi) = R(\theta, \phi^*) \geq \min_{\phi \in \widehat{\Phi}} R(\theta, \phi) \geq \min_{\phi_1, \ldots, \phi_m \in \Phi} \min_{\phi \in \{\phi_i\}_{i=1}^m} R(\theta, \phi)$. This is contradicting to the fact that *(iii)* holds. Hence, the Lemma is proved. □

Table 3: Performance of ROSE and baselines under various disturbances in **Hopper environment.**

| Method | ROSE-all | ROSE-worst |
|---|---|---|
| Hopper (No disturbance) | 0.86±0.07 | **0.95±0.01** |
| Hopper(Action noise) | 0.81±0.01 | **0.91±0.006** |
| Hopper (Worst Adversary) | 0.63±0.22 | **0.84±0.14** |

## C  ABLATION STUDIES FOR UNDERSTANDING THE COST OF OVER-PESSIMISM

To validate our theory in Section 3, we conduct extra experiments in the Hopper environment. We investigate two versions of ROSE that updates the adversarial head with different schemes: in each iteration during training, *(i)* ROSE-worst: only update the worst-$k$ adversaries, where the worst-$k$ adversaries are defined as in Section 3.2, and *(ii)* ROSE-all: update the whole population of adversaries.

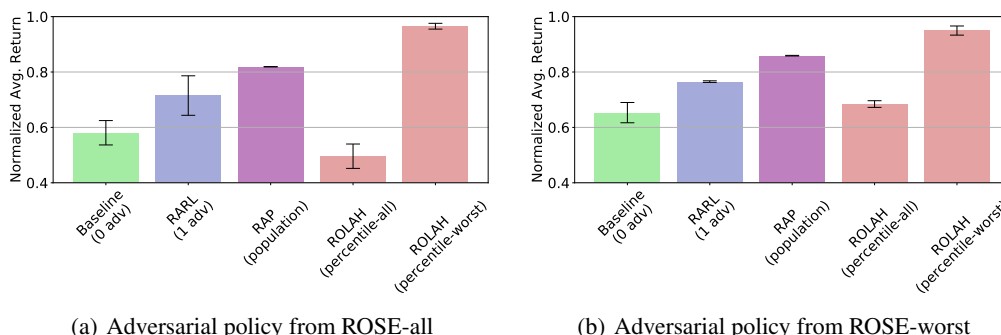

(a) Adversarial policy from ROSE-all       (b) Adversarial policy from ROSE-worst

Figure 4: Average normalized return for Hopper task cross-tested with the worst adversary from (1) ROSE-all and (2) ROSE-worst.

## C.1 ROBUSTNESS TO DISTURBANCE ON THE AGENT

We report the normalized return of ROSE with different update methods as the discussion in Section 4 in Table 3 for 3 types of disturbances during evaluation: (*i*) no disturbance, (*ii*) random adversary that adds noise to the actions of the agents, and (*iii*) the worst adversary that represents the worst case performance of a given policy. Empirical evidence demonstrates that ROSE (referred to as *ROSE-all*) leads to more robustness to disturbance compared with ROSE-all, which supports our analysis in Section 3.

## C.2 ROBUSTNESS TO TEST CONDITIONS (ENVIRONMENTAL CHANGES)

We follow the same evaluation metrics as we demonstrate in Section 4, considering training with a fixed pair of mass and friction values while evaluating the trained policies with varying mass and friction coefficients. We show that the ability to generalization is better with only updating the worst$-k$ adversaries during training in Figure 3.

## C.3 CROSS-VALIDATION OF ROSE

After the training process of ROSE is finished, we have access to a trained agent and a group of trained adversarial policies. To evaluate the effectiveness of training, we evaluate all the baseline methods and ROSE-worst/all under the disturbance from two adversaries: *(i)* the worst adversary in the trained adversarial ensemble of ROSE-worst and *(ii)* the worst adversary in the trained adversarial ensemble of ROSE-all. The selection of the worst adversary follows the same process as described in Section 4. As can be seen in Figure 4, ROSE-all cannot survive from its own adversary, i.e., the adversary that it has encountered during training.

## D ABLATION STUDIES ON THE RL ALGORITHM IMPLEMENTING ROSE

In Section 4, we adopt TRPO as our core baseline and consider different adversarial algorithms built on top of TRPO. Here we mainly conduct the experiments on the Hopper, Half-Cheetah, and Walker2d tasks using Proximal Policy Optimization (PPO) (Schulman et al., 2017a) and show the robustness comparison with varying test conditions in Figure 5 and with various disturbances in Table 4. We also extend our method (ROSE) to an off-policy version using DDPG (Lillicrap et al., 2019), demonstrating better performance consistently in Table 5. Our ROSE performs better against other baselines using PPO and DDPG, indicating that our approach is not limited to a specific RL policy optimization method.

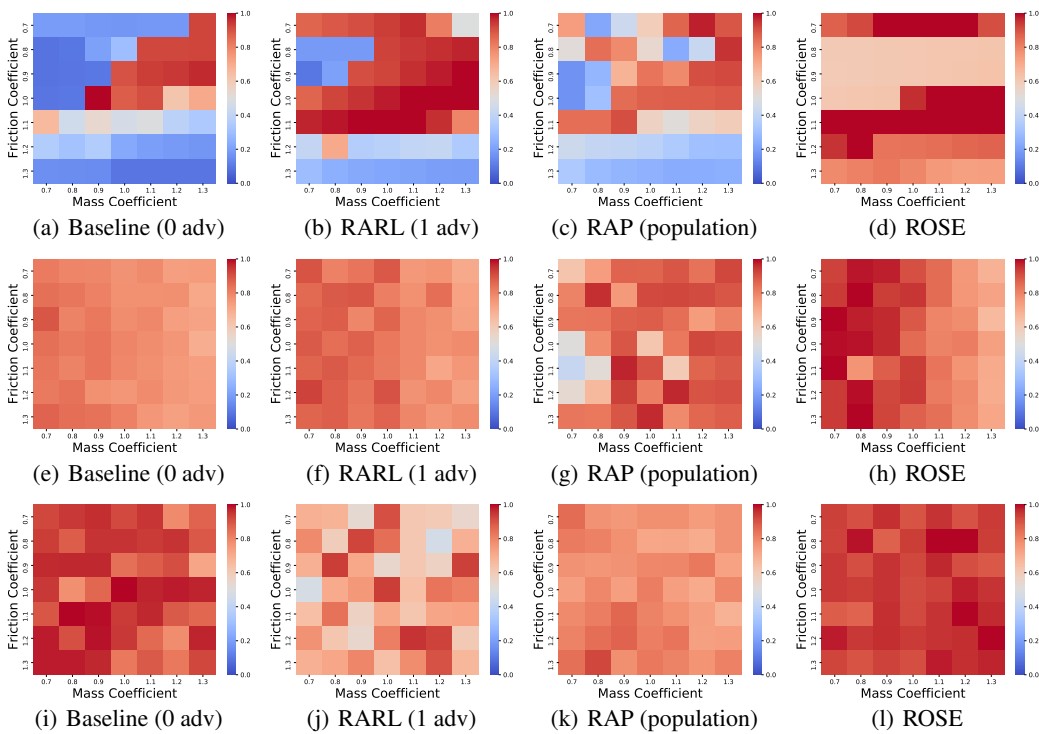

Figure 5: Average normalized return across 10 seeds tested via different mass coefficients for **PPO** on the x-axis and friction coefficients on the y-axis. High reward has red color; low reward has blue color. $1^{st}$ row: Hopper, $2^{nd}$ row: Half-Cheetah, $3^{rd}$: Walker2d

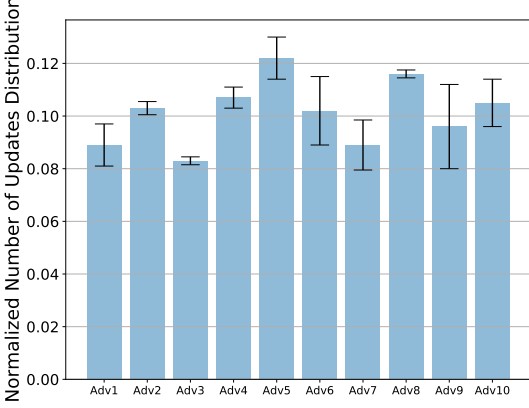

Figure 6: Number of updates for the adversaries in Ant environment with N=10 and k=3

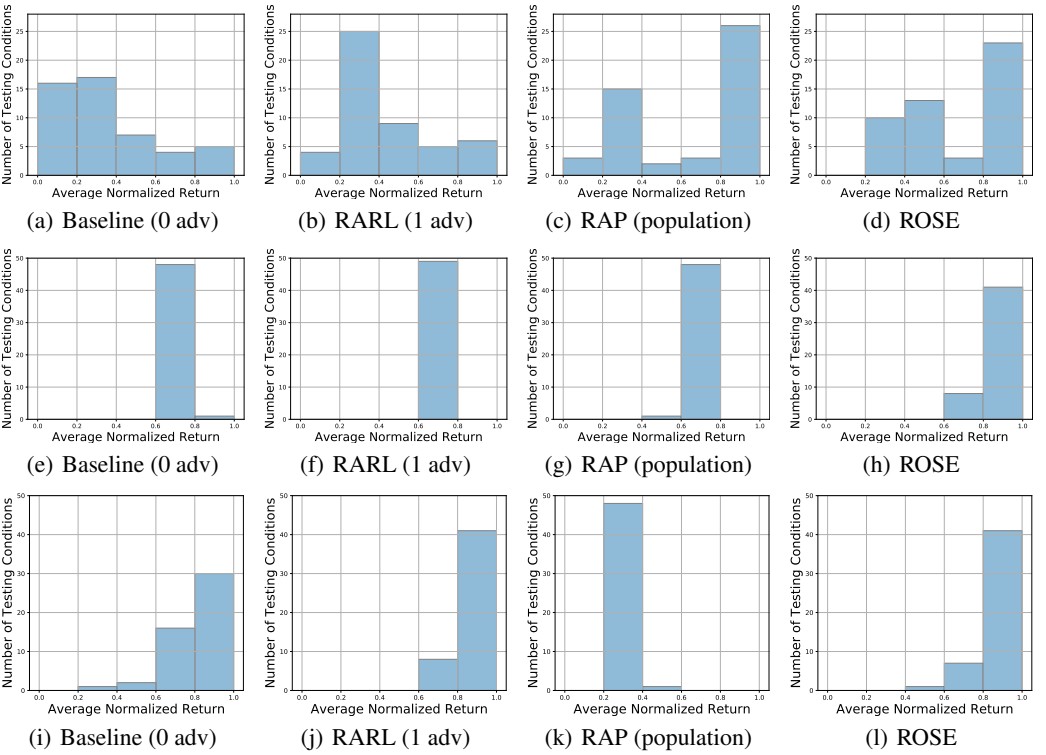

Figure 7: Distribution of average normalized return across 10 seeds with jointly varying test conditions. High reward on the right; low reward on the left. $1^{st}$ row: Hopper, $2^{nd}$ row: Half-Cheetah, $3^{rd}$: Walker2d.

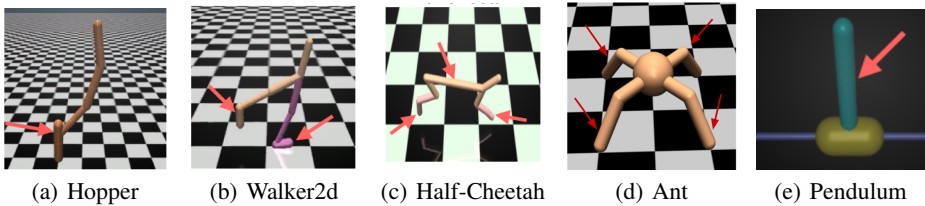

Figure 8: Illustrations of the environments evaluated in our experiments.

Table 4: Performance of ROSE and baselines under various disturbances for **PPO**.

| Method | Baseline (0 adv) | RARL (1 adv) | RAP (population adv) | ROSE |
|---|---|---|---|---|
| Hopper (No disturbance) | 0.89±0.009 | **0.97±0.003** | 0.87±0.003 | **0.97±0.33** |
| Hopper(Action noise) | 0.72±0.07 | **0.94±0.002** | 0.53±0.2 | 0.88±0.001 |
| Hopper (Worst Adversary) | 0.65±0.04 | **0.88±0.009** | 0.68±0.17 | 0.87±0.24 |
| Half-Cheetah (No disturbance) | 0.89±0.04 | 0.91±0.04 | 0.89±0.08 | **0.92±0.08** |
| Half-Cheetah(Action noise) | **0.91±0.03** | 0.89±0.10 | 0.53±0.43 | **0.91±0.03** |
| Half-Cheetah (Worst Adversary) | 0.21±0.24 | 0.24±0.04 | 0.28±0.39 | **0.51±0.43** |
| Walker2d (No disturbance) | 0.94±0.32 | 0.91±0.33 | 0.90±0.10 | **0.98±0.09** |
| Walker2d (Action noise) | 0.93±0.29 | 0.86±0.35 | **0.99±0.14** | 0.98±0.03 |
| Walker2d (Worst Adversary) | 0.30±0.13 | 0.51±0.16 | 0.53±0.24 | **0.71±0.37** |

Table 5: Performance of ROSE and baselines under various disturbances using **DDPG** with Ant environments

| Method | Baseline (0 adv) | RARL (1 adv) | RAP (population adv) | ROSE |
|---|---|---|---|---|
| Ant (No disturbance) | 0.80±0.12 | 0.84±0.06 | 0.86±0.09 | **0.89±0.10** |
| Ant (Action noise) | 0.58±0.19 | 0.61±0.18 | **0.63±0.12** | 0.63±0.15 |
| Ant (Worst Adversary) | 0.20±0.07 | 0.26±0.09 | 0.28±0.15 | **0.35±0.14** |

## E EXPERIMENTAL DETAILS

All our experiments are run on Nvidia RTX A5000 with 24GB RAM and our implementation are partly based on the codes published by *rllab* (Duan et al., 2016). In our experiments, 10 adversarial candidates are considered in RAP and ROSE and select the worst-$k$ adversaries for updating in each iteration with $k = 3$. We implement our method as well as existing baselines using TRPO and PPO. We list the hyperparameters we choose in Table 6. The clipping range for PPO is $0.2$. For those hyperparameters which are not listed, we adopt the default values in *rllab*.

Table 6: The hyperparameter used for experiments.

| Hyperparameters | Values |
|---|---|
| No of layers | 3 |
| Neurons in each layer | 256, 256, 256 |
| Batch Size | 4000 |
| Discount Factor ($\gamma$) | 0.995 |
| GAE parameter ($\lambda$) | 0.97 |
| No of iterations | 500 |

