# OpenReview forum: "Robust Reinforcement Learning with Structured Adversarial Ensemble"
_ICLR.cc/2024/Conference — Submitted to ICLR 2024_

### Official Review · Reviewer_APuc · 2023-10-31

**Soundness:** 3 good
**Presentation:** 3 good
**Contribution:** 4 excellent
**Rating:** 6
**Confidence:** 4

**Summary:**

This paper solves the "robust RL" problem by noting two major difficulties - over-optimism and over-pessimism - in the previous state-of-the-art. The robust RL considered in this paper is adversarial training using a two-player max-min game. This paper provides justifications for the proposed approach and evaluate extensively on benchmarks.

**Strengths:**

I think this work really pushes the adversarial RL community research efforts further by answering:

> can we design computationally efficient adversarial RL algorithms which are not pessimistic to unrealistic adversaries?

The main contribution of game-theoretic algorithm with solutions to two concerns (over-optimism and over-pessimism) raised in the adversarial problem setting is a really nice idea worthy for publication. My score reflect the weaknesses.

**Weaknesses:**

I have only a few weakness for this work as follows:

- In summary, this approach is finding appropriate number of adversaries (resolving over-optimism) to do a domain randomization (DR) step (averaging over worst-k adversaries) on top of these many adversaries. I see this paper mentions Mehta et al 2020 in related work but due credit to more DR works are missing. I am not sure of any DR-related works, but i'll appreciate if the authors can include more related works on these methodologies and discuss their differences with the current approach.

- Related work need to be better than combining all different settings into one paragraph:
> Subsequent works generalize the objective to unknown uncertainty sets, and formulate the uncertainty as perturbations/disturbance introduced into, or intrinsically inherited from, the environments, including perturbations in the constants which are used to define environmental dynamics (e.g., gravity, friction, mass) (Abraham et al., 2020; Mankowitz et al., 2019; Mehta et al., 2020; Pinto et al., 2017; Vinitsky et al., 2020a; Tessler et al., 2019; Vinitsky et al., 2020a;b), disturbance introduced to the observations (Zhang et al., 2020) and actions (Li et al., 2021; Tessler et al., 2019).

-- The current framework considers robustness against adversarial actions. Tessler et al., (2019) and thereafter are the closest to current work's setting. Of course then their algorithms need to be benchmarked against.

-- Transition model perturbation can be justified in the framework mentioning the evolution of the environment depends on the adversarial actions. Model uncertainty in robust RL is defined in more generality [2-10]. So it will be better to include more detailed Related Works including [2-10] and more relevant works in the revision. I agree this work includes experiments with model uncertainty, and the baseline is M2TD3 that fits into [2-10] line of works. There also exists works on state uncertainty [1].

To summarize, major update on the related works is required. If published as is, due credit to appropriate works will be missed and misrepresented.
Side note: I've also stopped at '10' since you get the idea of inadequate related work discussion.

I am open to discussions with the authors and reviewers to make sure the work quality matches the score, which I believe so at this point, but a potential reach to 8 definitely exists. All the best for future decisions!

[1] Robust Multi-Agent Reinforcement Learning with State Uncertainty Sihong He, Songyang Han, Sanbao Su, Shuo Han, Shaofeng Zou, and Fei Miao. Transactions on Machine Learning Research, June 2023.

[2] Xu. Z, Panaganti. K, Kalathil. D, Improved Sample Complexity Bounds for Distributionally Robust Reinforcement Learning. Artificial Intelligence and Statistics, 2023.

[3] Nilim, A. and El Ghaoui, L. (2005). Robust control of Markov decision processes with uncertain transition matrices. Operations Research, 53(5):780–798

[4] Iyengar, G. N. (2005). Robust dynamic programming. Mathematics of Operations Research, 30(2):257–280.

[5] Panaganti, K. and Kalathil, D. (2021). Robust reinforcement learning using least squares policy iteration with provable performance guarantees. In Proceedings of the 38th International Conference on Machine Learning, pages 511–520.

[6] Panaganti, K. and Kalathil, D. (2022). Sample complexity of robust reinforcement learning with a generative model. In Proceedings of The 25th International Conference on Artificial Intelligence and Statistics, pages 9582–9602.

[7] Roy, A., Xu, H., and Pokutta, S. (2017). Reinforcement learning under model mismatch. In Advances in Neural Information Processing Systems, pages 3043–3052.

[8] Panaganti, K., Xu, Z., Kalathil, D., and Ghavamzadeh, M. (2022). Robust reinforcement learning using offline data. Advances in Neural Information Processing Systems (NeurIPS).

[9] Shi, L. and Chi, Y. (2022). Distributionally robust model-based offline reinforcement learning with near-optimal sample complexity. arXiv preprint arXiv:2208.05767

[10] L Shi, G Li, Y Wei, Y Chen, M Geist, Y Chi (2023) The Curious Price of Distributional Robustness in Reinforcement Learning with a Generative Model, NeurIPS 2023

**Questions:**

-na-

---

> ### Author Response · Authors · 2023-11-16
>
> Dear Reviewer APuc,
>
> We really appreciate your detailed review. Please see below our responses to your concerns.
>
> > *Related Works.*
>
> Thank you so much for referring us to these additional references.  Following your advice, we have updated the manuscript to include a more thorough discussion of the related works. For your convenience, we replicate the updated Related Work section (only the paragraphs about robust RL) below:
>
> One research topic closely related to our method is domain randomization. Domain randomization is a technique to increase the generalization capability over a set of pre-defined environments. The set of environments are parameterized (e.g., friction and mass coefficient) to allow the agent to encode the knowledge about the deviations between training and testing scenarios. The environment parameters are commonly uniformly sampled during training [19,  21, 22, 23]. Even though ADR [20] is proposed to learn a parameter sampling strategy on the top of domain randomization, all of the aforementioned methods are not learned over the worst case scenarios. Moreover, in real life application, if not chosen carefully, the environment set can also lead to over-pessimism with a larger range while selecting a smaller range of the set will be over-optimistic. Hence, our proposed method can be readily extended into domain randomization by considering the environments as fixed adversaries.
>
> Robustness to transition models has been widely investigated. It was initially studied by robust MDPs [2,3,4] through a model-based manner by assuming the uncertainty set of environmental transitions is known, which can be solved by dynamic programming. In this approach, a base dynamic model is assumed and the uncertainty set is crafted as a ball centered around the base model with a predetermined statistical distance or divergence, e.g., KL-divergence or Wasserstein distance. Following works address scenarios where the base model is unknown but samples from the base model are available.  For example,  [6,10] propose model-based algorithms that first estimates the base model and then solve the robust MDP; [5,7] propose online model-free policy evaluation and policy iteration algorithms for robust RL with convergence guarantees; [1] proposes algorithms with polynomial guarantees for tabular cases where both the number of states and actions are finite.; [8,9] further extends the study of robust RL with only offline data. In contrast to these works, we follow the approach of RARL which does not explicitly specify the set of environments but learns a robust policy by competing with an adversary.
>
> Subsequent works generalize the objective to unknown uncertainty sets, and formulate the uncertainty as perturbations/disturbance introduced into the environments [10,11, 12, 13,14]. Notably, RARL [14] introduces an adversary with the objective to affect the environment to minimize the agent’s rewards. Notably, while in this work we focus on robustness to the transition model, there are two other types of robustness: robustness to the disturbance of actions [15,16] and robustness to state/observation [17,18].
>
> > *The current framework considers robustness against adversarial actions. Tessler et al., (2019) and thereafter are the closest to current work's setting. Of course then their algorithms need to be benchmarked against.*
>
> Thank you for this suggestion. It seems our work considers a rather different setup than Tessler et al. (2019), and hope our response here could help clarify the potential misunderstanding here. Specifically, our work pursues the goal similar to the RARL work, i.e., **toward the robustness against perturbed environment by adversarial training (as part of our methodology), which would be rather different than solving the problem of robustness against adversarial actions provided by the other party (what Tessler et al. (2019) considered)**. In the framework of RARL, the adversary conducts actions to change the environment to minimize the agent’s rewards. The action space of the adversarial is often different to that of the agent while Tessler et al., (2019)  assumes that the protagonist and the adversary share the same action space, which is distinct to the experimental setup in RARL and our work.
>
> *Chen Tessler, Yonathan Efroni, and Shie Mannor. Action robust reinforcement learning and applications in continuous control. In International Conference on Machine Learning, pp. 6215–6224. PMLR, 2019.*

---

> ### Author Response · Authors · 2023-11-16
>
> [1] Zaiyan Xu, Kishan Panaganti, and Dileep Kalathil. Improved Sample Complexity Bounds for Distributionally Robust Reinforcement Learning. International Conference on Artificial Intelligence and Statistics, 2023
>
> [2] Iyengar, G. N. (2005). Robust dynamic programming. Mathematics of Operations Research, 30(2):257–280.
>
> [3] J Andrew Bagnell, Andrew Y Ng, and Jeff G Schneider. Solving uncertain markov decision processes. Citeseer, 2001.
>
> [4] Arnab Nilim and Laurent Ghaoui. Robustness in markov decision problems with uncertain transition matrices. Advances in neural information processing systems, 16, 2003.
>
> [5] Panaganti, K. and Kalathil, D. (2021). Robust reinforcement learning using least squares policy iteration with provable performance guarantees. In Proceedings of the 38th International Conference on Machine Learning, pages 511–520.
>
> [6] Panaganti, K. and Kalathil, D. (2022). Sample complexity of robust reinforcement learning with a generative model. In Proceedings of The 25th International Conference on Artificial Intelligence and Statistics, pages 9582–9602.
>
> [7] Roy, A., Xu, H., and Pokutta, S. (2017). Reinforcement learning under model mismatch. In Advances in Neural Information Processing Systems, pages 3043–3052.
>
> [8] Panaganti, K., Xu, Z., Kalathil, D., and Ghavamzadeh, M. (2022). Robust reinforcement learning using offline data. Advances in Neural Information Processing Systems (NeurIPS).
>
> [9] Shi, L. and Chi, Y. (2022). Distributionally robust model-based offline reinforcement learning with near-optimal sample complexity. arXiv preprint arXiv:2208.05767
>
> [10] ​​L Shi, G Li, Y Wei, Y Chen, M Geist, Y Chi (2023) The Curious Price of Distributional Robustness in Reinforcement Learning with a Generative Model, NeurIPS 2023
>
> [11] Ian Abraham, Ankur Handa, Nathan Ratliff, Kendall Lowrey, Todd D Murphey, and Dieter Fox. Model-based generalization under parameter uncertainty using path integral control. IEEE Robotics and Automation Letters, 5(2):2864–2871, 2020.
>
> [12]Daniel J Mankowitz, Nir Levine, Rae Jeong, Yuanyuan Shi, Jackie Kay, Abbas Abdolmaleki, Jost Tobias Springenberg, Timothy Mann, Todd Hester, and Martin Riedmiller. Robust reinforcement learning for continuous control with model misspecification. arXiv preprint arXiv:1906.07516, 2019.
>
> [13]Eugene Vinitsky, Yuqing Du, Kanaad Parvate, Kathy Jang, Pieter Abbeel, and Alexandre Bayen. Robust reinforcement learning using adversarial populations, 2020a.
>
> [14]Lerrel Pinto, James Davidson, Rahul Sukthankar, and Abhinav Gupta. Robust adversarial reinforcement learning. In International Conference on Machine Learning, 2017.
>
> [15] Chen Tessler, Yonathan Efroni, and Shie Mannor. Action robust reinforcement learning and applications in continuous control. In International Conference on Machine Learning, pp. 6215–6224. PMLR, 2019.
>
> [16] Yutong Li, Nan Li, H Eric Tseng, Anouck Girard, Dimitar Filev, and Ilya Kolmanovsky. Safe reinforcement learning using robust action governor. In Learning for Dynamics and Control, pp. 1093–1104. PMLR, 2021.
>
> [17] Huan Zhang, Hongge Chen, Chaowei Xiao, Bo Li, Mingyan Liu, Duane Boning, and Cho-Jui Hsieh. Robust deep reinforcement learning against adversarial perturbations on state observations. Advances in Neural Information Processing Systems, 33:21024–21037, 2020.
>
> [18] Robust Multi-Agent Reinforcement Learning with State Uncertainty Sihong He, Songyang Han, Sanbao Su, Shuo Han, Shaofeng Zou, and Fei Miao. Transactions on Machine Learning Research, June 2023.
>
> [19] Josh Tobin, Rachel Fong, Alex Ray, Jonas Schneider, Wojciech Zaremba, and Pieter Abbeel. Domain randomization for transferring deep neural networks from simulation to the real world. Intelligent Robots and Systems (IROS), 2017 IEEE/RSJ International Conference on. IEEE, 2017.
>
> [20] Bhairav Mehta, Manfred Diaz, Florian Golemo, Christopher J Pal, and Liam Paull. Active domain randomization. Conference on Robot Learning, pp. 1162–1176. PMLR, 2020.
>
> [21] X. B. Peng, M. Andrychowicz, W. Zaremba, and P. Abbeel, “Sim-to-Real Transfer of Robotic Control with Dynamics Randomization,” in 2018 IEEE international conference on robotics and automation (ICRA), pp. 3803–3810, IEEE, 2018.
>
> [22]  Z. Li, X. Cheng, X. B. Peng, P. Abbeel, S. Levine, G. Berseth, and K. Sreenath, “Reinforcement learning for robust parameterized locomotion control of bipedal robots,” in 2021 IEEE International Conference on Robotics and Automation (ICRA), pp. 2811–2817, IEEE, 2021.
>
> [23] J. Siekmann, Y. Godse, A. Fern, and J. Hurst, “Sim-to-real learning of all common bipedal gaits via periodic reward composition,” in 2021 IEEE International Conference on Robotics and Automation (ICRA), pp. 7309–7315, IEEE, 2021.

---

> ### Author Response · Authors · 2023-11-20
> **Gentle Reminder**
>
> Dear Reviewer APuc,
>
> As the discussion period is closing to an end, we would like to know if there are any other concerns that we can help resolve. The authors are more than happy and fully committed to providing further clarification and addressing any remaining questions you may have. If our responses have effectively addressed all of your primary concerns, we sincerely ask the reviewer to kindly consider an upward adjustment of the score to reflect the improvements made.
>
> Thank you once again for the precious time and effort that you have invested in reviewing this work.
>
> Best wishes,
>
> Authors

---

> ### Comment · Reviewer_APuc · 2023-11-21
> **Ack**
>
> Thank you for the detailed response. I think the authors have improved the manuscript compared to the pre-rebuttal stage but there are concerns from other reviewers that need further iterations. I will update my score after the author-reviewer discussion period.

---

> > ### Author Response · Authors · 2023-11-22
> > **Thank you**
> >
> > Dear reviewer APuc,
> >
> > The authors sincerely appreciate your constructive suggestions and comments. It is the dedication from reviewers like you that establishes the standing of ICLR.
> >
> > Best,
> >
> > Authors

---

### Official Review · Reviewer_c2QC · 2023-11-01

**Soundness:** 3 good
**Presentation:** 3 good
**Contribution:** 2 fair
**Rating:** 3
**Confidence:** 5

**Summary:**

Deep reinforcement learning (RL) has demonstrated its capability to generate optimal strategies for environments with intricate dynamics. However, there are inherent challenges: the vastness of the parameter search space and limited exploration during training can compromise the robustness and performance guarantees of resulting policies. One technique that bolsters the resilience of RL agents is robustness through adversarial training. In this method, a hostile agent (adversary) aims to minimize the RL agent's cumulative reward by causing disturbances in the environment. Although this framework has strengths, two primary issues emerge: over-optimism due to difficulties in solving inner optimization problems, and over-pessimism from broad, imprecise candidate adversary sets which may consider unrealistic disturbance scenarios. To address these challenges, this study introduces a structured adversarial ensemble where multiple adversaries operate concurrently. This ensemble approach both improves the estimation of worst-case scenarios and shifts the RL agent's objective from absolute worst-case performance to an average of the most challenging scenarios. The proposed method outperforms existing robust RL strategies, and experiments show that it consistently enhances robustness across different environmental disturbances.

**Strengths:**

This paper is well-written and easy-to-follow.

**Weaknesses:**

* Limit technical novelty. RARL with an adversarial population is not a novel idea. The difference between ROSE and RAP is very incremental.
* The theorems and lemmas fall short of providing insights into the theoretical justification of the algorithm design, for example, why to optimize the performance over the worst-$k$ adversaries, how to choose the $k$ value, etc.
* One important assumption is that the adversaries are distinct enough from each other. However, there is no component in the algorithm that aims to improve diversity explicitly, such as in ADR (Bhairav Mehta et al., 2020). Therefore, I don't think it would be considered a 'structured ensemble'.
* Missing related work:
    * Shen, Macheng, and Jonathan P. How. "Robust opponent modeling via adversarial ensemble reinforcement learning." Proceedings of the International Conference on Automated Planning and Scheduling. Vol. 31. 2021.
    * Huang, Peide, et al. "Robust reinforcement learning as a Stackelberg game via adaptively-regularized adversarial training." Proceedings of the Thirty-First International Joint Conference on Artificial Intelligence (IJCAI-22).
    * Zhai, Peng, et al. "Robust adaptive ensemble adversary reinforcement learning." IEEE Robotics and Automation Letters 7.4 (2022): 12562-12568.

**Questions:**

See weakness.

---

> ### Author Response · Authors · 2023-11-16
>
> Dear Reviewer c2QC,
>
> We greatly appreciate your comments. We have found quite a few discrepancies between the comments from the reviewer about specific parts of our methodology, and what had been introduced in the paper -- we have enriched them with more details below, and hopefully they could help clarify the misunderstandings/confusions.
>
> ---
>
> > *Limit technical novelty. RARL with an adversarial population is not a novel idea.*
>
> Compared to other works, our work has the following contributions:
>
> 1. We **identified and addressed two major bottlenecks at once toward robust RL through adversarial training**, the potential over-optimism and over-pessimism issues. As a result, our method, as demonstrated by our extensive empirical study (e.g., see Figure 2 and Table 1), significantly improves RARL and RAP which fail to address these problems, demonstrating the cruciality of the identified problems.
> 2. Our work, to the best of our knowledge, is the first that provides thorough **theoretical understanding** to illustrate the benefit of introducing ensembles to robust RL, which **laid out the theoretical ground that future works can follow**. Moreover, our method differs from the classic average/vote-based ensemble methods, as it considers only a (self-adapted) moving subset of the ensemble that are most helpful for combating potential over-optimism/pessimism.
> 3. We conducted extensive experiments and ablation studies to verify the efficacy of our method. The results have shown that our method consistently dominates baselines across **different environments, with various types of disturbance applied, over a variety (i.e., on-policy and off-policy) of backbone RL algorithms.**
>
> While the amount of difference between algorithms is a challenging question to settle, the difference of our proposed method to other ensemble method-based algorithms has proven to lead to significant performance improvement. This exactly underscores the technical importance of our proposed method. As the other two reviewers are satisfied with the novelty of our method, we would greatly appreciate it if the reviewer could further pinpoint from which perspectives we could further improve the novelty of our work, in order to lead to your satisfaction.
>
> > *Connection between theorem and the algorithm*
>
> We have detailed the insights brought by the theoretical results in the paragraph **“Insights from the Theoretical Results”** in Section 3.1. To reiterate, our theoretical results have shown that the number of the adversaries required is of a moderate polynomial order. There are two parameters to choose: the total number of adversaries and the k-value. With the analysis corroborating that the total number of adversaries is a relatively small number and the fact that k-value is upper bounded by the total number of adversaries, the search space is significantly reduced by optimizing over the worst-k adversaries (which does not necessarily compensate for optimality either).
>
> > *Why to optimize the performance over the worst-k adversarial, how to choose the k-value, etc.*
>
> As presented in Section 3.2, we propose to optimize over the worst-k adversaries to address the potential over-pessimism caused by the misspecification of the adversary set – among other things, this is a novelty of our work that enables our  method to significantly outperform existing (state-of-the-art) methods. In most practical real-world scenarios, it is often challenging to have a precise characterization of the adversary set, thus leaving the robust RL algorithms with an adversary set with agents mostly leading to irrelevant and infeasible outcomes, e.g., an adversary that applies extreme disturbance that is unfeasible to occur in realistic scenarios. The effectiveness of our methodology is concretely justified by extensive experiments in Section 4, because the objective of the protagonist can be significantly diverted if it optimizes against solely the worst adversary who may apply unrealistic disturbances. Hence, the choice of k-value depends on to what extent one would want to minimize the effects brought in by such unrealistic adversaries. In principle, larger $k$-value should be used for environments with straightforward dynamics, where the adversaries are more likely to discover extreme strategies easily.
>
> > *However, there is no component in the algorithm that aims to improve diversity explicitly*
>
> Thank you for this question. By Occam’s Razor, we believe it is beneficial to retain only necessary components of an algorithm. **As illustrated by empirical studies ( Section 6 Ablation A4), we have confirmed that as long as the adversaries are initialized differently, they will remain distinct.** And the consistently improved performance has demonstrated that the distinction from random initialization is sufficient. Hence, we choose to not induce extra components that can potentially cause stability issues and consume more computation complexity.

---

> > ### Author Response · Authors · 2023-11-16
> > **Continuation**
> >
> > > “Missing related work”
> >
> > Thank you for referring us to these works. We have included them in our updated manuscript.
> >
> > On the other hand, please note that **some of these works consider rather distinct set-ups to ours**. Specifically, [1] focus on the scenario where there are other agents with unknown objectives and employs an ensemble to simulate the behaviors of these agents but not for a policy with robustness to environmental disturbance; [2] employs Stackelberg game to address the potential over-conservatism in scenarios where the adversaries do not act simultaneously, while our work follows the conventional Nash equilibrium widely employed by the robust RL works [4]; [3] proposes to adaptively scale the weights of of a set of adversaries to improve stability and robustness, while our method employs the ensemble differently to address the over-pessimism caused by potential misspecification of the adversary set. Moreover, our work additionally establishes theoretical support and rigorous understanding for the application of ensemble methods in robust RL, which is the element missing in these works.
> >
> > ---
> >
> > [1] Shen, Macheng, and Jonathan P. How. "Robust opponent modeling via adversarial ensemble reinforcement learning." Proceedings of the International Conference on Automated Planning and Scheduling. Vol. 31. 2021
> >
> > [2] Huang, Peide, et al. "Robust reinforcement learning as a Stackelberg game via adaptively-regularized adversarial training." Proceedings of the Thirty-First International Joint Conference on Artificial Intelligence (IJCAI-22).
> >
> > [3] Zhai, Peng, et al. "Robust adaptive ensemble adversary reinforcement learning." IEEE Robotics and Automation Letters 7.4 (2022): 12562-12568.
> >
> > [4] Moos, Janosch, et al. Robust Reinforcement Learning: A Review of Foundations and Recent Advances. Mach. Learn. Knowl. Extr. 2022, 4, 276-315.

---

> ### Author Response · Authors · 2023-11-20
> **Gentle Reminder**
>
> Dear Reviewer c2QC,
>
> As the discussion period is closing to an end, we would like to know if there are any other concerns that we can help resolve. The authors are more than happy and fully committed to providing further clarification and addressing any remaining questions you may have. If our responses have effectively addressed all of your primary concerns, we sincerely ask the reviewer to kindly consider an upward adjustment of the score to reflect the improvements made.
>
> Thank you once again for the precious time and effort that you have invested in reviewing this work.
>
> Best wishes,
>
> Authors

---

> > ### Comment · Reviewer_c2QC · 2023-11-22
> > **Re**
> >
> > Thanks to the authors for the response. However, my original concerns regarding novelty, the main theorem, and the algorithm design still remain.
> >  - Two major bottlenecks, namely over-optimism and over-pessimism, have been identified by numerous literature in robust RL, as evidenced by the literature review. Therefore, it is hard to count them as novelties.
> >  - The disjoint between the theorem and empirical results. As the authors pointed out in the **Insights from the Theoretical Results**, "Lemma 3 implies that the true benefit brought by the adversarial ensemble lies in the optimization process instead of the final optimal solution it offers." In this case, why can a theorem about the approximation error and the final optimal solution justify the empirical results if the true benefit is brought by the optimization process? Maybe the authors could provide more insight into this justification.
> >  - The authors claim that "as illustrated by empirical studies (Section 6 Ablation A4), as long as the adversaries are initialized differently, they will remain distinct". There are two comments regarding this.
> >    - First, ablation A4 only shows the adversaries get updated evenly and does not actually say anything about the diversity of the adversary. For example, let's say the ensemble has no diversity, e.g., output $10 \pm \epsilon, \epsilon \sim \mathcal N (0, 0.1)$ every iteration. They will get updated evenly, but it has little to do with the diversity.
> >    - Second, maintaining diversity is not an "unnecessary component." As shown by a number of works such as Stein Variation Policy Gradient [1] and Active Domain Randomization [2], diversity is not going to be ensured just by initializing differently; different initialization could lead to very similar state-action visiting distributions. It is heavily dependent on the optimization landscape as well as the choice of hyperparameters.
> >
> > Combining all the aforementioned reasons, I think the current version does not warrant acceptance. I encourage the authors to improve the manuscripts and resubmit.
> >
> > **Ref:**
> > [1] Liu, Yang, et al. "Stein variational policy gradient." arXiv preprint arXiv:1704.02399 (2017).
> > [2] Mehta, Bhairav, et al. "Active domain randomization." Conference on Robot Learning. PMLR, 2020.

---

> > > ### Author Response · Authors · 2023-11-22
> > > **Further Clarification**
> > >
> > > Dear Reviewer c2QC,
> > >
> > > Thank you for your response. However, it appears that the raised comments are orthogonal to the one posted in the initial review. With that being said, the authors are more than happy to clarify these, and we sincerely hope these scholarly exchanges can resolve the reviewer’s misunderstanding.
> > >
> > >
> > > > “Two major bottlenecks have been identified by numerous literature in robust RL, as evidenced by the literature review.”
> > >
> > > We would like to note that this work focuses on the problems of over-optimism and over-pessimism specific to **adversarial training toward robust RL**. While there exist works targeting the selection problem of uncertainty set in robust MDP-motivated approaches that explicitly define sets of environment dynamics, to the best of our knowledge, there hasn’t been any work with an identical objective to ours. We would appreciate if the authors can refer us to the works that also identifies and addresses the **over-optimism and over-pessimism for adversarial training** as the reviewer did not provide any reference. Moreover, as pointed out in our original response to the novelty concern, identifying such a problem is only one part of the novelty of this work.
> > >
> > > > “The disjoint between the theorem and empirical results…Maybe the authors could provide more insight into this justification.”
> > >
> > > While the reviewer’s original comment was about “theoretical justification of the algorithm design”, this follow-up comment seems to fall within a rather different aspect. **Our theoretical analysis provides a rigorous understanding of the benefits of ensemble methods in robust RL and establishes that the cost (i.e., number of required models in the ensemble) is moderate.** Theorem 1 and 2 imply that an ensemble can help provide a better approximated solution, thus leading to improved performance. This justifies the outstanding empirical results. On the other hand, the significantly improved performance of our algorithm helps justify the correctness of our analysis.
> > >
> > > In regards to Lemma 3, think of two functions $f_1(x)$ and $f_2(x)$ with identical optimal values at identical stationary points $x^*$ while $f_1$ is smooth and convex while $f_2$ is irregular with discontinuities on a set with non-zero measure. Although they have the **same solution**, $f_1(x)$ is much easier to optimize/approximate. In other words, ensemble transforms the original problem into an easier one while keeping the optimal value the same.
> > >
> > > > Diversity of the adversaries.
> > >
> > > **The authors are deeply confused by the reviewer’s comparison of our work to [1,2].** To be more concrete, the algorithms proposed in [1,2] are **drastically distinct** to ours: [1] does not address robust RL and does not involve adversarial training at all while [2] bases on [1] and randomly samples from a set of pre-defined environmental parameters, without a worst-case performance objective. As they focus on rather distinct problems and setups, their algorithms differ significantly to ours. The authors are baffled by the claim that the observation of not enough diversity in some hardly related methods imply that it is a universal problem. Since the optimal amount of required diversity across adversaries is challenging to presume in practice, it is only natural to require **enough** diversity for good performance. Given that there have been extensive experiments demonstrating that the diversity within the ensemble ensured by our method already leads to significantly improved performance, **We believe that we have sufficiently justified the diversity of adversaries empirically**, and sincerely hope that these scholarly exchanges help clarify the reviewer’s misunderstanding.
> > >
> > > ---
> > >
> > > [1] Liu, Yang, et al. "Stein variational policy gradient." arXiv preprint arXiv:1704.02399 (2017).
> > >
> > > [2] Mehta, Bhairav, et al. "Active domain randomization." Conference on Robot Learning. PMLR, 2020.

---

> > > > ### Comment · Reviewer_c2QC · 2023-11-22
> > > > **Re:**
> > > >
> > > > Thanks for the further response. However, some of my concerns are still not addressed adequately. Let me provide the references and reiterate some questions I have:
> > > >
> > > > > List of references for over-optimism and over-pessimism for adversarial training in RL.
> > > >
> > > > - **Overfitting to the worst case**:
> > > >     - Vinitsky, Eugene, et al. "Robust reinforcement learning using adversarial populations." arXiv preprint arXiv:2008.01825 (2020).
> > > >     - Gleave, Adam, et al. "Adversarial policies: Attacking deep reinforcement learning." arXiv preprint arXiv:1905.10615 (2019).
> > > >     - Dennis, Michael, et al. "Emergent complexity and zero-shot transfer via unsupervised environment design." Advances in neural information processing systems 33 (2020): 13049-13061.
> > > > - **Adversarial training with worst $\epsilon$ percentile of returns to optimize the conditional value at risk (CVaR)**
> > > >     - Rajeswaran, Aravind, et al. "Epopt: Learning robust neural network policies using model ensembles." arXiv preprint arXiv:1610.01283 (2016).
> > > >     - Chow, Yinlam, et al. "Risk-sensitive and robust decision-making: a cvar optimization approach." Advances in neural information processing systems 28 (2015).
> > > >
> > > > This is a non-comprehensive list of literature. For a complete survey, please refer to Moos, Janosch, et al. "Robust reinforcement learning: A review of foundations and recent advances." Machine Learning and Knowledge Extraction 4.1 (2022): 276-315.
> > > >
> > > >
> > > > > Reference to SVPG and ADR.
> > > >
> > > > Without explicit regularization, there is no extensive empirical or theoretical evidence for wishing ANY particle-based optimization to converge to distinct solutions just by initializing the particle differently. Actually, SVPG and the extensions from it show counter-examples for that (Figure 2 of SVPG). Again, ablation A4 offers no direct evidence for the diversity the authors claimed. Instead of the updating frequency, showing the state-action visiting distribution might be more helpful.

---

> > > > > ### Author Response · Authors · 2023-11-23
> > > > >
> > > > > Dear reviewer c2QC,
> > > > >
> > > > > While the gesture to continue discussion is appreciated and the authors are more than happy to respond to questions closely connected to our work, we would like to note that **bringing up related works that are already discussed in our manuscript and some other irrelevant works with drastically distinct problems of focus and settings and without any explanation does not help establish the claims** and would be detrimental to the review process. With all the best wishes, please find below our response.
> > > > >
> > > > > > Referred Works
> > > > >
> > > > > We have indeed cited and compared with [1] which, as discussed in the original submission, can be considered as a special case of our algorithm by setting $k$ equal to the total number of adversaries (the maximum possible value of $k$). Motivated by our analysis, this implies that **the algorithm in [1] does not spend enough effort on the worst-case optimization objective. The significantly improved performance of our algorithm also helps corroborate this.** Moreover, our work provides a rigorous understanding of the benefits of ensemble methods in robust RL which also benefits [1].
> > > > >
> > > > > Again we would like to note that this work addresses the problems of over-optimism and over-pessimism specific to **adversarial training toward robust RL**, a popular and effective robust RL approach. [2,3,4,5] have orthogonal problems of focus and settings. Specifically, instead of focusing on robust RL, [2] investigates the weakness of RL policies toward an adversarial policy, i.e. how a simple adversary can significantly disturb the reward of the policy. There is no content about “overfitting to the worst case in a pre-specified adversary set” in [2].  [3] trains a policy to generate environmental parameters which is distinct to the setting of RARL. Hence, [3] does not follow the approach of RARL and thus does not address the over-pessimism in adversarial training for robust RL. [4] assumes access to multiple “source domains” and does not follow the adversarial training approach proposed in RARL. [5] defines a new type of MDP (CVaR MDP) to replace the robust MDP setting which is the conventional framework for robust RL and focus of this work. In fact, [4,5] were published even before RARL.
> > > > >
> > > > > Going back to the reviewer's original concern on novelty that our work has been covered by existing ones -- the reviewer did not pinpoint which works have covered the topic we emphasize on (i.e, which works not cited by our work also addresses the problems of over-optimism and over-pessimism specific to adversarial training toward robust RL), but rather bringing up orthogonal works that focus on other aspects of robust RL than us. Moreover, **throwing out a survey paper but failed to identify any works that covered our topic rather justifies the novelty of our work.** To this end, the authors cannot help but doubt the validity of the original comment questioning our novelty.
> > > > >
> > > > > > Comments about Particle-based Optimization
> > > > >
> > > > > While the reviewer has demonstrated an outstanding interest in the convergence behavior of particle-based methods, we would like to underscore again that the convergence to considerately distinct adversaries is **never the focus of this work**, and we have never claimed that we have addressed this problem. The key of the success of our proposed algorithm is that the ensemble only needs to maintain **enough diversity for improved performance**. This is already extensively demonstrated by our empirical studies. The authors is baffled by the reviewer’s excessive focus on the properties of particle-based methods, and believe it is against the principle of a fair evaluation that although **extensive evidences have already been provided to demonstrate the effectiveness of the proposed approach on the target problem** (the problems of over-optimism and over-pessimism involved in the adversarial training toward robust RL), the contribution of our work is still devalued simply because it has not solved another unrelated problem.
> > > > >
> > > > > Best,
> > > > >
> > > > > Authors
> > > > >
> > > > > ---
> > > > >
> > > > > [1] Vinitsky, Eugene, et al. "Robust reinforcement learning using adversarial populations." arXiv preprint arXiv:2008.01825 (2020).
> > > > >
> > > > > [2] Gleave, Adam, et al. "Adversarial policies: Attacking deep reinforcement learning." arXiv preprint arXiv:1905.10615 (2019).
> > > > >
> > > > > [3] Dennis, Michael, et al. "Emergent complexity and zero-shot transfer via unsupervised environment design." Advances in neural information processing systems 33 (2020): 13049-13061.
> > > > >
> > > > > [4] Rajeswaran, Aravind, et al. "Epopt: Learning robust neural network policies using model ensembles." arXiv preprint arXiv:1610.01283 (2016).
> > > > >
> > > > > [5] Chow, Yinlam, et al. "Risk-sensitive and robust decision-making: a cvar optimization approach." Advances in neural information processing systems 28 (2015).

---

### Official Review · Reviewer_1Lc9 · 2023-11-02

**Soundness:** 3 good
**Presentation:** 3 good
**Contribution:** 3 good
**Rating:** 6
**Confidence:** 4

**Summary:**

The paper examines the limitations of reinforcement learning (RL) in robust policy design due to potential environmental disturbances. It identifies two key issues in adversarial training for RL: over-optimism from complex inner optimization and over-pessimism from the selection of adversarial scenarios. The authors propose an adversarial ensemble approach to address over-optimism and optimize average performance against the worst-k adversaries to mitigate over-pessimism. The theoretical underpinnings of this method are presented, and its efficacy is demonstrated through comprehensive experimental results.

**Strengths:**

1. **Structure and Clarity**: The paper is well-structured and provides clear explanations, enhancing readability and comprehension.

2. **Clear Motivation**: The authors articulate the significance of addressing both over-optimism and over-pessimism in adversarial training, which effectively establishes the paper's purpose.

3. **Theoretical Foundation**: The paper offers a solid theoretical analysis, bolstering the credibility of the proposed method.

4. **Experimental Validation**: The inclusion of extensive experimental results substantiates the claims and demonstrates the practical benefits of the proposed algorithm.

**Weaknesses:**

1. **Figure Clarity**: Figure 1, intended to aid in understanding, is unclear. A more straightforward illustration with an improved caption is needed.

2. **Explanation of Solution to Over-Pessimism**: The rationale behind using the average performance over the worst-k adversaries, as presented in Section 3.2, requires further clarification to be convincing. Can the authors elaborate on how the average performance optimization directly counteracts over-pessimism?

Overall, in my opinion, the paper contributes a thoughtful approach to improving the robustness of RL algorithms, which is substantiated by both theoretical analysis and experimental results. Despite some ambiguity in graphical representation and the need for additional explanations in certain sections, the paper is a good candidate for ICLR. Clarity improvements in the mentioned areas could enhance the paper's impact.

**Questions:**

Please refer to weaknesses.

---

> ### Author Response · Authors · 2023-11-16
> **Improved Clarify**
>
> Dear Reviewer 1Lc9,
>
> Thank you so much for your dedicated effort in helping improve our work! Please see below our detailed responses.
>
> ---
>
> > *Figure 1. A more straightforward illustration with an improved caption is needed.*
>
> We appreciate this advice and we have updated the manuscript to include a new Figure 1 with improved caption.
>
> > *Can the authors elaborate on how the average performance optimization directly counteracts over-pessimism?*
>
> Thank you for this insightful question. In most practical real-world scenarios, it is often challenging to have a precise characterization of the adversary set, thus leaving the robust RL algorithms with an adversary set with agents mostly leading to irrelevant and infeasible outcomes, e.g., an adversary that applies extreme disturbance that is unfeasible to occur in realistic scenarios. Hence, we propose to optimize over worst-k adversaries so that **robust RL can prevent the over-pessimism without precise knowledge of the adversary set**. This approach is highly effective, as demonstrated by the compelling results from extensive experiments, because the objective of the protagonist can be significantly diverted if it optimizes against solely the worst adversary who may unfortunately be one of the erroneously chosen adversaries in the adversary set.
>
> Moreover, as the max-min problems in robust RL are normally solved by iterative updates of the protagonist and the adversaries, where in each iteration we have an adversary $\phi$ against whom we will optimize the protagonist. However, if the adversary set is not precise, $\phi$ may be a misspecified scenario. If the rest $k-1$ adversaries (or the majority of the worst-$k$ adversaries) are indeed in the true interested scenarios, optimizing the average over the worst-k adversaries distracts the attention of the protagonist from the single uninterested worst case to the cases of interest. However, we cannot directly optimize the average performance of all the adversaries because this can lead to too little consideration of the worst-case performance, as demonstrated by our empirical study in Figure 2.
>
> We find this comment very helpful for improving the clarity on the motivations of this work, and we have already updated our manuscript to include this detailed explanation (in red text).

---

> ### Author Response · Authors · 2023-11-20
> **Gentle Reminder**
>
> Dear Reviewer 1Lc9,
>
> As the discussion period is closing to an end, we would like to know if there are any other concerns that we can help resolve. The authors are more than happy and fully committed to providing further clarification and addressing any remaining questions you may have. If our responses have effectively addressed all of your primary concerns, we sincerely ask the reviewer to kindly consider an upward adjustment of the score to reflect the improvements made.
>
> Thank you once again for the precious time and effort that you have invested in reviewing this work.
>
> Best wishes,
>
> Authors

---

### Author Response · Authors · 2023-11-16
**Summary of Revision**

Dear Reviewers and Area Chair,

We sincerely appreciate your effort in helping improve our work. In the initial review, two (out of three) reviewers (1Lc9, APuc) are impressed by both the algorithmic novelty and the theoretical contribution of our work, and it is glad to see that all three reviewers unanimously agree that the manuscript is well-written. The reviewers have shared insightful comments that are greatly helpful for improving the clarity over specific parts of our presentation. Following these suggestions, we have updated the manuscript from the following aspects, i.e.,

1. Updated Figure 1 that now introduces the motivation visually.
2. Added more thorough discussions on related works.
3. Provided more details on how our method can help address the over-pessimism problem.

We hope such changes would make our work more concrete. Moreover, we would appreciate any follow-ups from the reviewers and are more than happy to respond to them as well.

Best,

Authors

---

### Author Response · Authors · 2023-11-23

Dear Chairs and Reviewers,

We express our gratitude again for your precious help in the review of our work. We have received very constructive comments from most reviewers and have updated our manuscript to echo them accordingly. However, **the authors must convey severe concern about the comments and the demonstrated prejudice from reviewer c2QC who currently holds distinct but rather unjustified opinions**.

Specifically, reviewer c2QC gave orthogonal comments against the ones from the initial review. **The 2 follow-ups from c2QC proposes significantly different comments and evidence to the original comments -- this rather indicated that concerns from the initial review have been resolved.** The discussion concentrates around two questions being the novelty and the need for a diversity component in our method. Our responses to these points are summarized below for the chairs and other reviewers to evaluate.

Toward the novelty, the reviewer c2QC initially claimed that the topic our work tries to resolve has been already looked into by much existing research. However, as discussed in our responses, all those works brought up by c2QC either chase for different objectives (i.e., not for robust RL) or follow unrelated setups (i.e., require multiple source domains or need to generate environments with distinct parameters), where some even have already been discussed in the related work section pointing out the distinctions (see our latest two responses to c2QC). We have also read the survey paper shared by c2QC and did not find any work covering our contributions, that are (i) to provide theoretical insights for the benefits of introducing ensemble method into robust RL, and (ii) introduce an algorithm to effectively address the over-optimism and over-pessimism with extensive experiments justifying the methodology in which our proposed methods have demonstrated **significantly improved performance** by addressing these two problems.

Besides, instead of straightforward yet effective solutions, reviewer c2QC appears to be biased toward overly complicated solutions, i.e., **reviewer c2QC insisted that our algorithm has to have a component promoting diversity of the adversarial ensemble even though our empirical results showed that sufficient diversity for outstanding empirical performance can be achieved without the need of such**. Reviewer c2QC also has demonstrated a strong interest in the convergence behavior of particle-based methods, which is orthogonal to the focus of this work. More generally, throughout the years, the community has benefited from both complicated and straightforward methodologies, for example, TRPO opened a new era in on-policy actor-critic learning while PPO significantly improved TRPO's performance with modifications. To this end, we would tend to believe that the solid theoretical analyses and strong empirical performance concretely support the motivation and effectiveness of our method.





We thank the chair and reviewers again for participating in discussions, allowing our work to be thoroughly evaluated. We believe that the chair and reviewers would consider opinions from both sides, and make a fair recommendation accordingly -- which is the inspiration that makes ICLR truly exceptional.

Best,

Authors

---

### Meta-Review · Area_Chair_ojWa · 2023-12-08

**Metareview:**

* Summary of the Paper:
The paper addresses the robustness of reinforcement learning (RL) in the face of environmental disturbances. It identifies two key issues in adversarial training for RL: over-optimism and over-pessimism. The authors propose an adversarial ensemble approach to address these challenges, theoretically substantiating their approach and empirically demonstrating its effectiveness.

* Reviewers’ Opinions:
	1.	Reviewer 1Lc9 praised the paper’s structure, clarity, and theoretical foundation. They pointed out the need for clearer explanations in certain areas, such as Figure 1 and the solution to over-pessimism.
	2.	Reviewer c2QC raised concerns about the technical novelty, theoretical justification, and the diversity component in the algorithm. They also mentioned missing related works and requested more robust evidence for the diversity of the adversary.
	3.	Reviewer APuc commended the paper’s contribution to adversarial RL and its game-theoretic algorithm. They suggested improvements in related work, particularly regarding domain randomization and robustness to transition models.

* Authors’ Responses:
The authors actively engaged with the reviewers, clarifying and updating their manuscript based on the feedback. They addressed concerns about figure clarity, the rationale behind their approach to over-pessimism, and elaborated on the novelty and theoretical aspects of their work. They also expanded the related work section and explained the alignment of their work with existing literature.

* Overall Assessment:
The concerns about novelty and technical depth, along with the need for improved clarity, were pivotal in the decision to recommend rejection.

**Justification For Why Not Higher Score:**

The AC, after a thorough examination of the paper, the reviews, and the authors’ responses, recommended rejection. This decision was influenced by the concerns raised by Reviewer c2QC regarding the paper’s novelty and technical aspects. Additionally, after reading the paper, the AC noted the need for further clarity in the paper.

The AC also acknowledged receiving a private message from the authors. The AC did not find any inappropriate behavior or unjustifiable reviews from reviewer c2QC.

**Justification For Why Not Lower Score:**

N/A

---

### Decision · Program_Chairs · 2024-01-16

Reject